# Recursive Neural Ordinary Differential Equations for Partially Observed Systems

## Abstract

Identifying spatiotemporal dynamics is a difficult task, especially in scenarios where latent states are partially observed and/or represent physical quantities. In this context, first-principle ordinary differential equation (ODE) systems are often designed to describe the system's dynamics. In this work, we address the problem of learning parts of the spatiotemporal dynamics with neural networks when only partial information about the system's state is available. Taking inspiration from recursive state estimation and Neural ODEs, we outline a general framework in which complex dynamics generated by differential equations with distinguishable states can be learned in a principled way. We demonstrate the performance of the proposed approach leveraging both numerical simulations and a real dataset extracted from an electro-mechanical positioning system. We show how the underlying equations fit into our formalism and demonstrate the improved performance of the proposed method when compared with standard baselines.

## 1 Introduction

Ordinary differential equations (ODEs) are used to describe the state evolution of many complex physical systems in engineering, biology, and other fields of natural sciences. Traditionally, first-principle notions are leveraged in designing ODEs as a form to impose physical meaning and interpretability (Psichogios & Ungar, 1992) of latent states. A major issue, however, is the inherent complexity of real-world problems for which even carefully designed ODE systems cannot account for all aspects of the true underlying physical phenomenon (Karniadakis et al., 2021). Moreover, we often require prediction of systems whose dynamics are not fully understood or are partially unknown (Imbiriba et al., 2022).

In this context, Neural ODEs (NODEs) (Chen et al., 2018) emerged as a powerful tool for learning complex correlations directly from the data, where residual neural networks (NNs) are used to parameterize the hidden ODEs' states. Extensions of NODE were developed to improve learning speed (Xia et al., 2021; Massaroli et al., 2021) and learning longtime dependencies in irregularly sampled time series(Xia et al., 2021). A major challenge in learning NODEs arises when latent states of interest contribute indirectly to the observations. This is the case when an unobserved state (in the sense that it is not measured) influences an observed state. In this scenario, NODE's standard solutions, which are optimized using the adjoint method (Boltyanskiy et al., 1962), are compromised. Furthermore, NODE systems may have infinitely many solutions since parameters and unobserved states are estimated jointly. As a consequence, even when the model is capable of fitting the data, unobserved states cannot be accurately inferred without incorporating some kind of prior information in the model (Demirkaya et al., 2021). Recently, new hybrid strategies have focused on mixing first-principle models and NODEs to constrain the solution space and obtain meaningful estimations of missing states (Imbiriba et al., 2022; Demirkaya et al., 2021; Ghanem et al., 2021). Despite the lack of a clear formalization, in these works the authors were imposing some kind of distinguishability among states by adding known parts of the dynamics, resulting in hybrid first-principle data-driven models. Nevertheless, these works focus on state estimation using data-driven components to improve or augment existing dynamics but fail to learn global models and do not scale for large parameterized models.

In this paper, we propose a sequential optimization approach that at each time step solves an alternating optimization problem for learning system dynamics under partially observed states, when states

are distinguishable. The approach focuses on learning unknown dynamics from data where the state related to the unknown dynamics is unobserved. Since the dynamics is unknown, we assume it is described by parametric models such as NNs. The proposed solution leverages the relationship between many recursive state-space estimation procedures and Newton's method (Humpherys et al., 2012) to develop an efficient recursive NODE learning approach capable of sequentially learning states and model parameters. The benefit of the sequential strategy is twofold: (1) reduce the need for accurate initial conditions during training; (2) avoids simultaneous estimation of all states, making second-order optimization methods feasible. Furthermore, the proposed approach exploits the distinguishable property of states by designing an alternating optimization strategy with respect to states and parameters. The result is an interconnected sequential optimization procedure, where at each step model parameters and data are used to estimate latent states, and corrected latent states are used to update the model parameters in the current optimization step. Such alternating optimization approach improves the optimization of system parameters since it estimates unobserved hidden states and uses them in learning system parameters. In the case of RNODE, it also prevents vanishing gradients. Moreover, we define distinguishable latent variables and test the proposed *Recursive* NODE (RNODE) in hybrid scenarios where NNs replace parts of the ODE systems such that the distinguishability of latent variables is kept. Finally, as a side effect of the recursive paradigm adopted the proposed strategy can assimilate data and estimate initial conditions by leveraging its sequential state estimation framework over past data.

## 2 RELATED WORK

### 2.1 PARTIAL OBSERVATION

In the context of data-driven ODE designs, most learning frameworks assume that all states are observed in the sense that they are directly measured. This assumption does not reflect many real-world scenarios where a subset of the states are unobserved. GP-SSM is a well-established approach used for dynamic systems identification (McHutchon et al., 2015; Ialongo et al., 2019). GP-SSM can be adapted by introducing a recognition model that maps outputs to latent states to solve the problem of partial measurements (Eleftheriadis et al., 2017). Nevertheless, these methods do not scale well with large datasets and are limited to small trajectories(Doerr et al., 2018). Indeed, (Doerr et al., 2018) minimizes this problem by using stochastic gradient ELBO optimization on minibatches. However, GP-SSM-based methods avoid learning the vector field describing the latent states and instead directly learn a mapping from a history of past inputs and observations to the next observation.

Similar approaches to the recognition models have been used for Bayesian extensions of NODEs, where the NODE describes the dynamics of the latent state while the distribution of the initial latent variable given the observations and vice versa are approximated by encoder and decoder networks (Yildiz et al., 2019; Norcliffe et al., 2021). The encoder network, which links observations to latent state by a deterministic mapping or by approximating the conditional distribution, can also be a Recurrent Neural Network (RNN) (Rubanova et al., 2019; Kim et al., 2021; De Brouwer et al., 2019), or an autoencoder (Bakarji et al., 2023). Despite focusing on mapping observations to latent states with neural networks and autoencoders, these works were not demonstrated to learn parameterized models under partial observations. Moreover, this parameterized line of work of mapping observation to latent states suffers from undistinguishability problem since several latent inputs could lead to the same observation. Recently, sparse approaches such as (Bakarji et al., 2022) merged encoder networks to identify a parsimonious transformation of the hidden dynamics of partially observed latent states. Moreover, Nonlinear Observers and recognition models were combined with NODEs to learn dynamic model parameters from partial observations while enforcing physical knowledge in the latent space (Buisson-Fenet et al., 2022). Differently from the aforementioned methods, in this work, we propose a recursive alternating approach that uses alternating Newton updates to optimize a quadratic cost function with respect to states and model parameters. Furthermore, the proposed strategy provides a systematic way to estimate initial conditions from historical data.

### 2.2 SECOND ORDER NEWTON METHOD

Despite the efficiency and popularity of many stochastic gradient descent methods (Robbins & Monro, 1951; Duchi et al., 2011; Hinton et al., 2012; Kingma & Ba, 2014) for optimizing NNs, great efforts have been devoted to exploiting second-order Newton methods where Hessian information is

used, providing faster convergence (Martens & Grosse, 2015; Botev et al., 2017; Gower et al., 2016; Mokhtari & Ribeiro, 2014). When training neural networks, computing the inverse of the Hessian matrix can be extremely expensive (Goldfarb et al., 2020) or even intractable. To mitigate this issue, Quasi-Newton methods have been proposed to approximate the Hessian pre-conditioner matrix such as Shampoo algorithm (Gupta et al., 2018), which was extended in (Anil et al., 2020) to simplify blocks of the Hessian, and in (Gupta et al., 2018) to be used in variational inference second-order approaches (Peirson et al., 2022). Similarly, works in (Goldfarb et al., 2020; Byrd et al., 2016) focused on developing stochastic quasi-Newton algorithms for problems with large amounts of data. It was shown that recursive the extended Kalman filter can be viewed as Gauss-Newton method (Bell, 1994; Bertsekas, 1996). Moreover, Newton's method was used to derive recursive estimators for prediction and smoothing (Humpherys et al., 2012). In this paper, we develop a recursive Newton method that mitigates the problem of partial observations of latent states.

## 3    Model and Background

In this section, we describe our modeling assumptions, discuss the distinguishability of latent states, and present the time evolution of the resulting generative model.

### 3.1    Model

In this work, we focus on stochastic differential equations (SDE) as defined in (Øksendal & Øksendal, 2003) to describe the evolution of system parameters $\theta(t) \in \mathcal{P} \subset \mathbb{R}^{d_\theta}$, latent states $x(t) \in \mathcal{X} \subset \mathbb{R}^{d_x}$, and observations (or measurements) $y(t) \in \mathcal{Y} \subset \mathbb{R}^{d_y}$. The joint process can be described as:

$$\begin{aligned}
\dot{\theta}(t) &= g(\theta(t)) + \dot{\nu}(t) \\
\dot{x}(t) &= f(x(t), \theta(t), u(t)) + \dot{\epsilon}(t) \\
y(t) &= h(x(t)) + \zeta(t)
\end{aligned} \tag{1}$$

where $\nu(t), \epsilon(t)$ and $\zeta(t)$ are Wiener processes. $u(t) \in \mathcal{U} \subset \mathbb{R}^{d_u}$ is a vector of external inputs, and the functions $g : \mathcal{P} \to \mathcal{P}$, $f : \mathcal{X} \times \mathcal{P} \times \mathcal{U}$, and $h : \mathcal{X} \to \mathcal{Y}$ describe the system parameters, latent and observation processes, respectively. To describe the evolution of system parameters $\theta(t)$ and latent states $x(t)$ we consider the process in equation 1 to be first-order Markov process evolving over time $t$.

**The partial observation problem:** Ideally, states $x(t)$ would be directly observed, and thus appear as an element in $y(t)$. In practice, some of these states could influence $y(t)$ only indirectly by acting on other measurable states. That is when classical training fails. In this work, we are interested in learning the unknown dynamics governing unobserved states. Note that this scenario poses further challenges over the estimation process since the recovery of latent states can be compromised.

### 3.2    Distinguishability of nonlinear systems

The task of recovering latent states $x(t)$ from a sequence of observations and inputs $\mathcal{D}_N \triangleq \{u(0), y(0), \ldots, u(N-1), y(N-1)\}$ rests on our ability to distinguish two observations $h(x(t_a))$ and $h(x(t_b))$ from one another.

**Definition 3.1** *We say that a pair of latent variables $x(t_a)$ and $x(t_b)$ are distinguishable with respect to a control sequence $u(t) \in \mathcal{U} \subset \mathbb{R}^{d_u}$ if*

$$h(x(t_a)) \neq h(x(t_b)) \quad \forall x(t_a) \neq x(t_b) \tag{2}$$

Otherwise, we say that the pair is indistinguishable with respect to $u(t)$.

If under a control input $u(t)$, $h(x(t_a)) = h(x(t_b))$, then the state estimator cannot identify the true state $x$ since it can assume the true state to be $x(t_a)$ when it's $x(t_b)$ and vice versa. Since our procedure relies on finding latent states $x(t)$ given a control input $u(t)$ and observation $y(t)$ and uses it to identify the ODE system, by estimating the model parameters $\theta(t)$, estimating the wrong state $x(t)$ will result in finding the wrong model parameters, hence training will fail. A way to impose state distinguishability is to incorporate prior knowledge regarding the relationship of states focusing on achieving the properties stated in Definition 3.1.

### 3.3 GENERATIVE MODEL

In the continuous model presented in (1), a continuous-time description for the latent processes is assumed even though the observations are recorded at discrete time points. The time evolution of the states $x(t)$ can therefore be expressed as time integration of (1) using an off-the-shelf ODE solver:

$$x(t_i) = x(t_{i-1}) + \int_{t_{i-1}}^{t_i} f(x(t), u(t), \theta(t))dt + \int_{t_{i-1}}^{t_i} \frac{\partial \epsilon(t)}{\partial t} dt$$
$$= \text{ODESolve}(f, x(t_{i-1}), u(t_{i-1}), \theta(t_{i-1}), t_{i-1}, t_i) + \epsilon(t) \tag{3}$$

we define

$$f_o(x(t_{i-1}), u(t_{i-1}), \theta(t_{i-1})) = \text{ODESolve}(f, x(t_{i-1}), u(t_{i-1}), \theta(t_{i-1}), t_{i-1}, t_i) + \epsilon(t) \tag{4}$$

and

$$g_o(\theta(t_{i-1})) = \text{ODESolve}(g, \theta(t_{i-1}), t_{i-1}, t_i), \theta(t_{i-1}) + \nu(t) . \tag{5}$$

Based on the continuous model presented in (1) we present the time evolution of the latent states by the following generative model:

$$\theta(t_i) = g_o(\theta(t_{i-1})) + \nu(t)$$
$$x(t_i) = f_o(x(t_{i-1}), u(t_{i-1}), \theta(t_{i-1})) + \epsilon(t) \tag{6}$$
$$y(t_i) = h(x(t_i)) + \zeta(t) .$$

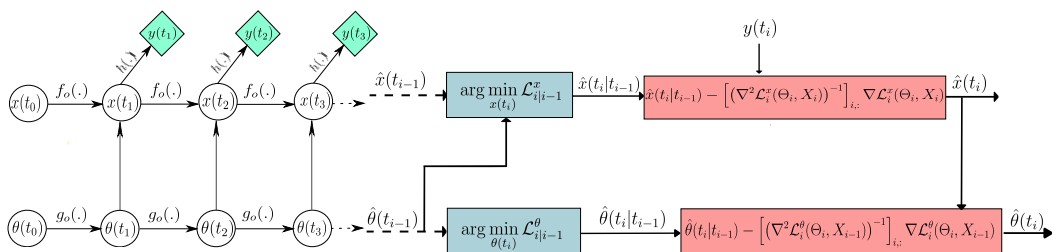

Figure 1: The generative model (left panel), and one step of RNODE (right panel).

## 4 METHOD

Recursive Neural Ordinary Differential Equations (RNODE) finds the model parameters $\theta(t)$ and latent states $x(t)$ given a dataset $\mathcal{D} \triangleq \{u(t_0), y(t_0), \ldots, u(t_{N-1}), y(t_{N-1})\}$ of discrete observations and control inputs when $x(t)$ is partially observed. Inspired by previous work describing the link between second-order Newton's method and the Kalman filter (Humpherys et al., 2012), the cost function $\mathcal{L}$ is updated and solved sequentially to find latent states $x(t)$ and model parameters $\theta(t)$ in one unified framework. RNODE assumes model distinguishability which implies that latent states $x(t)$ are recoverable from observations $y(t)$. In this context, we break the optimization steps into two concerning optimization with respect to $x(t)$ and $\theta(t)$.

### 4.1 SEQUENTIAL NEWTON DERIVATION

We denote by $\Theta_N = [\theta(t_0), \ldots, \theta(t_N)]$ and $X_N = [x(t_0), \ldots, x(t_N)]$ to be the set of latent states sampled at $t_0, t_1, \ldots, t_N$. To train the model, we optimize $(\Theta_N, X_N)$ to minimize a quadratic cost function starting from initial $\{x(t_0), \theta(t_0)\}$ using a collection of combined observation and input sequences $\mathcal{D}$ where the cost function is defined as:

$$\mathcal{L}_N(\Theta_N, X_N) = \frac{1}{2} \sum_{i=1}^{N} \|x(t_i) - f_o(x(t_{i-1}), u(t_{i-1}), \theta(t_{i-1}))\|_{Q_x^{-1}}^2$$
$$+ \|y(t_i) - h(x(t_i))\|_{R_y^{-1}}^2 + \|\theta(t_i) - g_o(\theta(t_{i-1}))\|_{Q_\theta^{-1}}^2 . \tag{7}$$

where $Q_x$, $R_y$ and $Q_\theta$ are known positive definite matrices, and $\|a - b\|^2_{A^{-1}} = (a - b)^T A^{-1}(a - b)$. As the Hessian's inverse is in general intractable, finding optimal solution $(\Theta^*_N, X^*_N)$ using the second order Newton method over the whole data set of size $N$ is unfeasible. For this reason, we resort to a sequential strategy by introducing a modified quadratic function $\mathcal{L}_i(\Theta_i, X_i)$. Let us re-write the cost function at time $t_i$ as:

$$\mathcal{L}_i(\Theta_i, X_i) = \mathcal{L}_{i-1}(\Theta_{i-1}, X_{i-1}) + \frac{1}{2}\|x(t_i) - f_o(x(t_{i-1}), u(t_{i-1}), \theta(t_{i-1}))\|^2_{Q_x^{-1}}$$
$$+ \frac{1}{2}\|y(t_i) - h(x(t_i))\|^2_{R_y^{-1}} + \frac{1}{2}\|\theta(t_i) - g_o(\theta(t_{i-1}))\|^2_{Q_\theta^{-1}} \tag{8}$$

where $\mathcal{L}_{i-1}(\Theta_{i-1}, X_{i-1})$ and $\mathcal{L}_i(\Theta_i, X_i)$ are the cost functions at times $t_{i-1}$ and $t_i$, respectively; $\Theta_i = [\theta(t_0), \dots, \theta(t_i)]$ and $X_i = [x(t_0), \dots, x(t_i)]$. In the sequential optimization paradigm, $\Theta_{i-1}$ and $X_{i-1}$ are assumed known and at the $i$-th optimization step is performed only with respect to $\{\theta(t_i), x(t_i)\}$. When $\{\theta(t_i), x(t_i)\}$ are determined jointly such as in (Humpherys et al., 2012), the optimization process will suffer from vanishing gradients under partial observations. However, if $x(t_i)$ is distinguishable, we can circumvent the vanishing gradient problem by first optimizing with respect to $x(t_i)$ and then $\theta(t_i)$. This will allow us to circumvent the partial observability problem and enable the use of an estimate of the unobserved state in training. To do so, we break the optimization function (8) into four alternating optimization procedures aiming at finding $\hat{x}(t_i)$ and then finding $\hat{\theta}(t_i)$ that minimizes (8) given $\hat{x}(t_i)$. Let us begin by defining two intermediate optimization functions $\mathcal{L}^x_{i|i-1}$ and $\mathcal{L}^\theta_{i|i-1}$ in (9) and (10) respectively as follows:

$$\mathcal{L}^x_{i|i-1}(\Theta_i, X_i) = \mathcal{L}_{i-1}(\Theta_{i-1}, X_{i-1}) + \frac{1}{2}\|x(t_i) - f_o(x(t_{i-1}), u(t_{i-1}), \theta(t_{i-1}))\|^2_{Q_x^{-1}}$$
$$+ \frac{1}{2}\|\theta(t_i) - g_o(\theta(t_{i-1}))\|^2_{Q_\theta^{-1}} \tag{9}$$

and

$$\mathcal{L}^\theta_{i|i-1}(\Theta_i, X_{i-1}) = \mathcal{L}_{i-1}(\Theta_{i-1}, X_{i-1}) + \frac{1}{2}\|\theta(t_i) - g_o(\theta(t_{i-1}))\|^2_{Q_\theta^{-1}}. \tag{10}$$

We proceed by optimizing (9) for $x(t_i)$ and (10) for $\theta(t_i)$, yielding the respective solutions below:

$$\hat{\theta}(t_i|t_{i-1}) = g_o(\hat{\theta}(t_{i-1}))$$
$$\hat{x}(t_i|t_{i-1}) = f_o(\hat{x}(t_{i-1}), \hat{\theta}(t_{i-1})). \tag{11}$$

Next, we define the two optimization functions responsible for the update steps for states and parameters. Specifically, we define $\mathcal{L}^x_i$ as:

$$\mathcal{L}^x_i(\Theta_i, X_i) = \mathcal{L}^x_{i|i-1}(\Theta_i, X_i) + \|y(t_i) - h(x(t_i))\|^2_{R_y^{-1}} \tag{12}$$

to be optimized with respect to $x(t_i)$ by minimizing $\mathcal{L}^x_i$ given intermediate values of equation (11) where:

$$\hat{x}(t_i) = \hat{x}(t_i|t_{i-1}) - \left[\left(\nabla^2 \mathcal{L}^x_i(\Theta_i, X_i)\right)^{-1}\right]_{i,:} \nabla \mathcal{L}^x_i(\Theta_i, X_i) \tag{13}$$

The solution to the problem above is given by given by (16). Equivalently, we define the update optimization function $\mathcal{L}^\theta_i$ as:

$$\mathcal{L}^\theta_i(\Theta_i, X_i) = \mathcal{L}^\theta_{i|i-1}(\Theta_i, X_{i-1}) + \|x(t_i) - f_o(x(t_{i-1}), u(t_{i-1}), \theta(t_{i-1}))\|^2_{Q_x^{-1}}$$
$$+ \|y(t_i) - h(x(t_i))\|^2_{R_y^{-1}} \tag{14}$$

to be optimized with respect to $\theta(t_i)$ by minimizing $\mathcal{L}^\theta_i$ given intermediate values of equation (11) and (16) as follows:

$$\hat{\theta}(t_i) = \hat{\theta}(t_i|t_{i-1}) - \left[\left(\nabla^2 \mathcal{L}^\theta_i(\Theta_i, X_{i-1})\right)^{-1}\right]_{i,:} \nabla \mathcal{L}^\theta_i(\Theta_i, X_{i-1}) \tag{15}$$

The resulting optimal variable $\hat{\theta}(t_i)$ is given by (17). The procedure is repeated until $t_i = t_N$. We present our main result in the following theorem:

**Theorem 4.1** *Given $\hat{\theta}(t_{i-1}) \in \hat{\Theta}_{i-1}$ and $\hat{x}(t_{i-1}) \in \hat{X}_{i-1}$, and known $P_{\theta_{i-1}} \in R^{d_\theta \times d_\theta}$ and $P_{x_{i-1}} \in R^{d_x \times d_x}$, the recursive equations for computing $\hat{x}(t_i)$ and $\hat{\theta}(t_i)$ that minimize (8) are given by the following :*

$$\hat{x}(t_i) = f_o(\hat{x}(t_{i-1}), \hat{\theta}(t_{i-1})) - P_{x_i}^- H_i^T (H_i P_{x_i}^- H_i^T + R_y)^{-1} \Big[ h\Big( f_o(\hat{x}(t_{i-1}), \hat{\theta}(t_{i-1})) \Big) - y(t_i) \Big] \tag{16}$$

$$\hat{\theta}(t_i) = g_o(\hat{\theta}(t_{i-1})) - G_{\theta_{i-1}} P_{\theta_i}^- F_{\theta_{i-1}}^T \Big[ f_o(\hat{x}(t_{i-1}), \hat{\theta}(t_{i-1})) - \hat{x}(t_i) \Big] \tag{17}$$

*with $P_{\theta_i}^-$, $P_{x_i}^-$ being intermediate matrices and $P_{\theta_i}$ and $P_{x_i}$ being the lower right blocks of $(\nabla^2 \mathcal{L}_i^\theta)^{-1}$ and $(\nabla^2 \mathcal{L}_i^x)^{-1}$ respectively:*

$$
\begin{aligned}
P_{\theta_i}^- &= P_{\theta_{i-1}} - P_{\theta_{i-1}} F_{\theta_{i-1}}^T \Big( Q_x + F_{\theta_{i-1}} P_{\theta_{i-1}} F_{\theta_{i-1}}^T \Big) F_{\theta_{i-1}} P_{\theta_{i-1}} \\
P_{x_i}^- &= F_{x_{i-1}} P_{x_{i-1}} F_{x_{i-1}} + Q_x \\
P_{x_i} &= P_{x_i}^- [I + H_i \left( R_y - H_i P_{x_i}^- H_i^T \right) H_i P_{x_i}^-] \\
P_{\theta_i} &= Q_\theta + G_{\theta_{i-1}} P_{\theta_i}^- G_{\theta_{i-1}}
\end{aligned}
\tag{18}
$$

*with $H_i, F_{x_{i-1}}, G_{\theta_{i-1}}$, and $F_{\theta_{i-1}}$ being the jacobians of the vector fields $h, f_o$ and $g_o$ at $\hat{x}(t_i|t_{i-1}), \hat{x}(t_{i-1})$ and $\hat{\theta}(t_{i-1})$ :*

$H_i = \frac{\partial h(\hat{x}(t_i|t_{i-1}))}{\partial \hat{x}(t_i|t_{i-1})}$, $F_{x_{i-1}} = \frac{\partial f_o(\hat{x}(t_{i-1}), \hat{\theta}(t_{i-1}))}{\partial \hat{x}(t_{i-1})}$, $F_{\theta_{i-1}} = \frac{\partial f_o(\hat{x}(t_{i-1}), \hat{\theta}(t_{i-1}))}{\partial \hat{\theta}(t_{i-1})}$ and $G_{\theta_{i-1}} = \frac{\partial g_o(\hat{\theta}(t_{i-1}))}{\partial \hat{\theta}(t_{i-1})}$.

The proof of Theorem 4.1 is provided in Appendix A.

As a consequence of Theorem (4.1), $\hat{x}(t_i)$ is computed according to (16) using $\hat{\theta}(t_{i-1})$. $\hat{\theta}(t_i)$ is computed afterwards according to (17) using $\hat{x}(t_i)$ that was previously found in (16). This alternating procedure between $x(t_i)$ and $\theta(t_i)$ is explained in the right panel of Figure 1, which depicts the four alternate optimization steps performed for each iteration $t_i$. The computational complexity of RNODE is detailed in Appendix D. An epoch of the RNODE has a complexity of $\mathcal{O}(N(d_x^3 + 2d_\theta^2 d_x + 2d_\theta d_x^2))$. Under the assumption that $d_\theta \gg d_x$ the complexity becomes $\mathcal{O}(N(2d_\theta^2 d_x + 2d_\theta d_x^2))$. During testing, however, the complexity becomes $\mathcal{O}(d_\theta)$ per step if integrating the learned mean vector field.

### 4.2 Obtaining initial condition from historical data

Obtaining initial conditions $x(t_0)$ during test time is often challenging. However, the proposed recursive framework can easily provide an estimate of the initial condition if historical data $\mathcal{D}_\mathcal{H} \triangleq \{u(t_{-N}), y(t_{-N}), \ldots, u(t_0), y(t_0)\}$ is available as described in equation 58 in Appendix C. Thus, given the model $\theta^*$ we can exploit the update equation for the states, see (17), to provide $\hat{x}(t_0)$.

## 5 Experiments

The performance of RNODE is assessed in comparison to state-of-the-art model learning methods on several challenging non-linear simulations and real-world datasets. We employed five different dynamical models to demonstrate the effectiveness of the proposed approach. For each dynamical model, we assumed that we don't have parts of the governing dynamics available, and replaced them with a neural network. In all of our experiments, we assume the latent process to be constant, that is $g(\theta(t) = 0$, since optimal $\theta(t)^*$ should be constant. Euler integrator is used as the ODE solver for efficiency and fast computation speed. Since the proposed mechanism rests on determining unobserved latent states from observed measurements, successful learning of the model relies on the distinguishability of latent states as defined in Definition (3.1). To ensure that, we assume partial knowledge of system ODE's.

As benchmark methods, we compared RNODE with three other well-established techniques for dynamical machine learning, namely NODE (Chen et al., 2018), RM (Buisson-Fenet et al., 2022)

and PR-SSM (Doerr et al., 2018). Currently, no code is available for the model learning frameworks presented in (Eleftheriadis et al., 2017). Moreover, the available code related to the works in (McHutchon et al., 2015; Ialongo et al., 2019) could be modified to account for the partial observation scenario. However, these algorithms become computationally unfeasible for medium and large datasets (Doerr et al., 2018). For that reason, we were not able to benchmark against these approaches. We emphasize that modifying the above-mentioned methods to either account for the ODE structure or make them computationally tractable is out of the scope of this paper. This also applies to the PRSSM method. Nevertheless, for the sake of providing comparative results, we still include results using PR-SSM which is computationally more efficient than other Gaussian process-based models but does not account for the ODE structure.

The benchmark results are summarized in Table 1 which represents normalized Root Mean Square Error (nRMSE) values for each model and method. In Figs. 2-5 we compare RM, PR-SSM, and our proposed method. All results were obtained with learned mean vector field integrated over time. Each subfigure represents the dynamics of a single state and contains ODE solutions for each method. We computed nRMSE using $\text{nRMSE} = \frac{\sqrt{\frac{1}{n}\sum_{i=1}^{n}(x(t_i)-\hat{x}(t_i))^2}}{\max(x(t))-\min(x(t))}$, where $\hat{x}(t_i)$ and $x(t_i)$ are the estimated and true states at time $t_i$, respectively, and $n$ is the number of data points.

Table 1: Comparison of nRMSE values for different dynamical models and methods.

| Methods | Neuron model | Yeast Glycolysis | Cart-pole | Harmonic Oscillator | EMPS |
|---|---|---|---|---|---|
| RM (Buisson-Fenet et al., 2022) | $2.39 \cdot 10^{-1}$ | $6.30 \cdot 10^{-1}$ | $1.06 \cdot 10^{0}$ | $2.36 \cdot 10^{-2}$ | $6.20 \cdot 10^{-1}$ |
| PR-SSM (Doerr et al., 2018) | $4.05 \cdot 10^{-1}$ | $1.59 \cdot 10^{0}$ | $1.52 \cdot 10^{0}$ | $1.21 \cdot 10^{0}$ | $4.05 \cdot 10^{1}$ |
| NODE (Chen et al., 2018) | $7.03 \cdot 10^{1}$ | $3.74 \cdot 10^{-1}$ | $2.84 \cdot 10^{-1}$ | $4.65 \cdot 10^{-1}$ | $1.65 \cdot 10^{0}$ |
| RNODE (Proposed) | $\mathbf{1.54 \cdot 10^{-1}}$ | $\mathbf{3.39 \cdot 10^{-2}}$ | $\mathbf{9.41 \cdot 10^{-3}}$ | $\mathbf{5.08 \cdot 10^{-3}}$ | $\mathbf{9.50 \cdot 10^{-2}}$ |

## 5.1 HODGKIN-HUXLEY NEURON MODEL

The renowned Hodgkin-Huxley Neuron Model (HH) (Hodgkin & Huxley, 1952) is an ODE system that describes the membrane dynamics of action potentials in neurons, which are electrical signals used by neurons to communicate with each other. The model has four states: $\dot{V}_m$ is the membrane potential, $n_{gate}$, $m_{gate}$, and $h_{gate}$ are gating variables controlling the membrane's ionic permeability. The equations governing the ODE system are provided in Eqs. 46-49 of the Appendix B.2. We train our recursive model with the assumption that

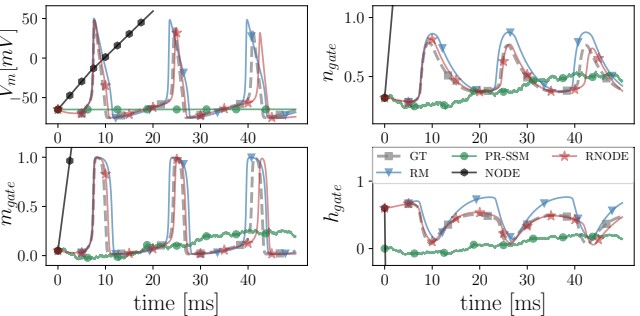

Figure 2: Learned state trajectories of HH model after training with RM, PR-SSM, NODE, and RNODE methods. Results are compared to ground truth ODE system trajectory labeled as GT. The proposed approach (RNODE) is capable of discerning the true trajectory for the unobserved state $h_{gate}$.

Eq. (49) governing dynamics of $h_{gate}$ is unknown and its corresponding state is not observed, i.e., $y(t_i) = (V_m(t_i), n_{gate}(t_i), m_{gate}(t_i))$. We replace the dynamics describing $\dot{h}_{gate}(t)$ by a neural network consisting of three layers. The first layer is a 20 units layer followed by an Exponential Linear Unit ($ELU$) activation function, second layer is also a 20 unit layer followed by a $tanh$ activation function. The last layer consists of 10 units with a $sigmoid$ activation function. We generate our dataset by applying a constant control input $u(t_i)$ to the HH model described in 46-49 for 50000 time steps with $dt = 10^{-3}s$ and by collecting measurements and inputs $\mathcal{D} \triangleq \{u(t_0), y(t_0), \ldots, u(t_{N-1}), y(t_{N-1})\}$. We train our model on $\mathcal{D}$ with $P_{x_0} = 10^{-2}I_{d_x}, P_{\theta_0} = 10^{2}I_{d_\theta} \ R_y = 10^{-10}I_{d_y}, Q_x = 10^{-5}I_{d_x}$ and $Q_\theta = 10^{-2}I_{d_\theta}$. At the beginning of each epoch, we solve the problem (58) of the Appendix C to get the initial condition. Final optimal parameters $\hat{\theta}(t_N)$ and initial condition $\hat{x}(t_0)$ are saved and collected at the end of

training. Fig. 2 depicts the dynamics of the system $\hat{\theta}(t_N)$ generated according to the generative model described in Eq (3) starting from initial condition $\hat{x}(t_0)$. The lower right panel demonstrates the superiority of the proposed model at learning $h_{gate}$.

To demonstrate the robustness of RNODE to different dynamical regimes and showcase its capability of estimating accurate initial conditions, we perform an additional experiment. For this, we generate data $\mathcal{D}_T$ with $N = 50,000$ samples using the HH model with different initial conditions from the ones used during training. From this data, we reserve the first 100 samples for learning the initial condition before performing integration for the remaining $49,900$ samples. Then, us-

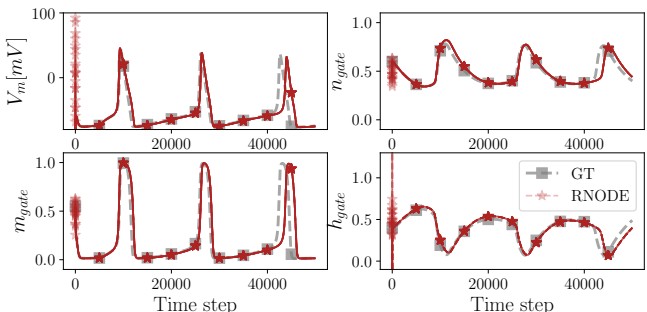

Figure 3: RNODE results for unknown initial condition. Initial conditions $\hat{x}(t_{100})$ were learned using the first 100 samples.

ing the learned model $\hat{\theta}(t_N)$ and the procedure described in Section 4.2 we obtained the initial condition $\hat{x}(t_{100})$ and obtained the RNODE solution. Figure 3 shows the evolution of the RNODE attesting to its capability of both estimating accurate initial conditions and generalization to other dynamical regimes.

## 5.2 CART-POLE SYSTEM

We demonstrate the efficacy of the proposed RNODE in learning the non-linear dynamics of the cart-pole system. The system is composed of a cart running on a track, with a freely swinging pendulum attached to it. The state of the system consists of the cart's position and velocity, and the pendulum's angle and angular velocity, while a control input $u$ can be applied to the cart. We used the LQR (Prasad et al., 2011) algorithm to learn a feedback controller that swings the pendulum and balances it in the inverted position in the middle of the

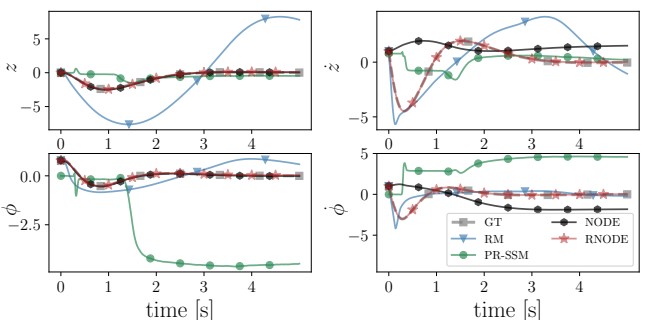

Figure 4: Learned state trajectories of the cart-pole system after training RM, PR-SSM, NODE, and RNODE methods. Results are compared to ground truth ODE system trajectory labeled as GT. We showed that the proposed approach (RNODE) is capable of discerning the true trajectory for the unobserved states $\dot{z}$ and $\dot{\phi}$.

track. The equations governing the ODE system are provided in Eqs (54)-(57) of the Appendix B.5.

We train our recursive model with the assumption that we don't know the equation corresponding to $\dot{\phi}$ governing dynamics of the cart-pole's angular rate. Therefore, we replace Eqs. (55) and (57) with a two-layer neural network with $tanh$ activation function on each layer. We don't measure cart-pole's velocity $\dot{z}(t_i)$ and angular rate $\dot{\phi}(t_i)$, i.e., $y(t_i) = [z(t_i), \phi(t_i)]$. We generate our dataset by applying LQR balancing controller to the cart-pole described in Eqs (54)-(57) for 5000 time steps with $dt = 10^{-3}s$ and by collecting measurements and inputs $\mathcal{D} \triangleq \{u(t_0), y(t_0), \ldots, u(t_{N-1}), y(t_{N-1})\}$. We train our model on $\mathcal{D}$ with $P_{x_0} = 10^{-2}I_{d_x}$, $P_{\theta_0} = 10^2 I_{d_\theta}$ $R_y = 10^{-10}I_{d_y}$, $Q_x = 10^{-5}I_{d_x}$ and $Q_\theta = 10^{-2}I_{d_\theta}$. At the beginning of each epoch, we solve problem (58) of the Appendix C to get the initial condition. Final optimal parameters $\hat{\theta}(t_N)$ and initial condition $\hat{x}(t_0)$ are saved and collected at the end of training We qualitatively assess the performance of our model by feeding the control sequence stored in $\mathcal{D}$ and parameters $\hat{\theta}(t_N)$ to the RNODE according to the generative model described in Eq (3) starting from initial condition $\hat{x}(t_0)$.

In Figure 4, we demonstrate the ability of the proposed RNODE to learn the underlying dynamics of the system partially observed data compared to RM and PR-SSM methods. Table 1 show that RNODE clearly outperforms the competing algorithms with nRMSE value that is 99.3% , 99.1%

and 97.67% smaller than the nRMSEs obtained by PR-SMM, RM, and NODE respectively. Analyzing the evolution of the latent states depicted in Figure 4, we notice that RNODE provides state trajectories that match the ground truth (GT) while the other two methods fail to capture the true trajectory. In fact, PR-SSM presents acceptable trajectories of $\dot{z}$ and $\dot{z}$ but fails to learn $\phi$ and $\dot{\phi}$ trajectories. On the other hand RM presents acceptable trajectories of $\phi$ and $\dot{\phi}$ but fails to learn $z$ and $\dot{z}$ trajectories. Moreover, the NODE successfully learns the observed $\phi$ and $z$ trajectories but fails to learn correct trajectories of the unobserved states $\dot{\phi}$ and $\dot{z}$. Both RM and PR-SSM estimated state trajectories are much more inaccurate than the one provided by RNODE. The main reason for this inaccuracy is that trajectory generation is run using a pre-computing control sequence $\mathcal{U} \triangleq \{u(t_0), \ldots, u(t_{N-1}))\} \in \mathcal{D}$, hence any inaccuracy in the learned dynamics would cause the trajectories to go way off the ground truth (GT) due to the nonlinearity of the cart-pole system. This shows the challenging nature of the problem and the proposed approach's efficiency in learning challenging nonlinear dynamics. In this context, RNODE's superior performance is due to its alternating optimization approach since estimates of unobserved states become available when optimizing $\theta$. This feature is unavailable in the competing methods.

### 5.3 Electro-mechanical positioning system

Here we evaluate the proposed RNODE on real data from an electro-mechanical positioning system described in (Janot et al., 2019). The training Dataset consists of system's of position, velocity, and control inputs used. The dataset consists of 24801 data points for each state and control input with $dt = 10^{-3}s$. In a similar fashion to the HH and cart-pole systems, we train the RNODE using position and control inputs. we replace the velocity's dynamics by a neural network of two layers of 50 and 20 units respectively followed by a $tanh$ activation function. Table 1 show that RNODE clearly outperforms the competing algorithms with nRMSE value that is 99.9% , 84.6% and 94.2% smaller smaller than the nRMSEs obtained by PR-SMM, RM, and NODE, respectively. Analyzing the evolution of the latent states depicted in Figure 5, we notice that RNODE provides state trajectories that match the ground truth (GT) while PR-SSM and RM collapse catastrophically. The NODE learns the period of the hidden $\dot{q}_m$ signal but

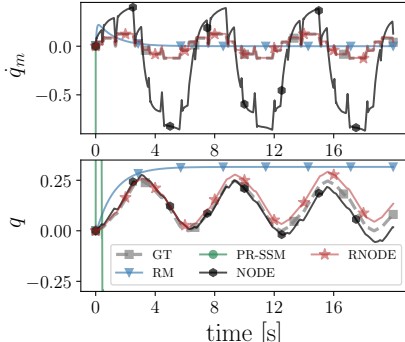

Figure 5: Learned state trajectories of EMPS after training RM, PR-SSM, NODE, and RNODE methods. Results are compared to ground truth ODE system trajectory labeled as GT. We showed that the proposed approach (RNODE) is capable of discerning the true trajectory for the unobserved state $\dot{q}_m$.

fails the capture its amplitude. The stiffness of $\dot{q}_m$ dynamics plays a role in these results since the sudden jumps shown in Figure 5 are hard to capture. This again demonstrates the robustness of the proposed approach.

## 6 Conclusions

We proposed a novel recursive learning mechanism for NODE's to address the challenging task of learning the complex dynamics of ODE systems with partial observations. Specifically, we constructed an alternating optimization procedure using Newton's method that sequentially finds optimal system latent states and model parameters. The resulting framework, RNODE, allows for efficient learning of missing ODEs when latent states are distinguishable. Different from other competing methods, RNODE optimizes model parameters using latent states instead of observed data, leading to superior performance under the partial observation setting. Experiments performed with three complex synthetic systems and one with real data provide evidence that RNODE is capable of providing adequate solutions in very challenging scenarios, attesting RNODE's superior performance when compared with other state-of-the-art strategies.

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

## A  PROOF OF THEOREM 1

Given a training dataset $\mathcal{D}_N \triangleq \{\boldsymbol{d}^{(0)}, \dots, \boldsymbol{d}^{(N-1)}\}$ with $\boldsymbol{d}^{(i)} = \{y(t_i), u(t_i)\}$, determine an optimal solution $(x^*(t_i), \theta^*(t_i))$ starting from known $(x(t_0), \theta(t_0), y(t_i))$ by solving the following mathematical optimization function:

$$
\begin{aligned}
\mathcal{L}_N(\Theta_N, X_N) = \frac{1}{N} \sum_{i=1}^{N} &\|x(t_i) - f_o(x(t_{i-1}), u(t_{i-1}), \theta(t_{i-1}))\|^2_{Q_x^{-1}} \\
&+ \|y(t_i) - h(x(t_i))\|^2_{R_y^{-1}} + \|\theta(t_i) - g_o(\theta(t_{i-1}))\|^2_{Q_\theta^{-1}}
\end{aligned}
\tag{19}
$$

Determining optimal solution $(\Theta_N^*, X_N^*)$ using the second order Newton method over the whole data set of size $N$ is computationally expensive. To solve this problem, we update the optimization function (19) as follows:

$$
\begin{aligned}
\mathcal{L}_i(\Theta_i, X_i) &= \mathcal{L}_{i-1}(\Theta_{i-1}, X_{i-1}) + \frac{1}{2}\|x(t_i) - f_o(x(t_{i-1}), u(t_{i-1}), \theta(t_{i-1}))\|^2_{Q_x^{-1}} \\
&\quad + \frac{1}{2}\|y(t_i) - h(x(t_i))\|^2_{R_y^{-1}} + \frac{1}{2}\|\theta(t_i) - g_o(\theta(t_{i-1}))\|^2_{Q_\theta^{-1}} \\
&= \mathcal{L}^x_{i|i-1}(\Theta_i, X_i) + \frac{1}{2}\|y(t_i) - h(x(t_i))\|^2_{R_y^{-1}} \\
&= \mathcal{L}^\theta_{i|i-1}(\Theta_i, X_{i-1}) + \frac{1}{2}\|x(t_i) - f_o(x(t_{i-1}), u(t_{i-1}), \theta(t_{i-1}))\|^2_{Q_x^{-1}} \\
&\quad + \frac{1}{2}\|y(t_i) - h(x(t_i))\|^2_{R_y^{-1}}
\end{aligned}
\tag{20}
$$

We start by minimizing the following equation with respect to $x$:

$$
\mathcal{L}_{i|i-1}^{x}(\Theta_i, X_i) = \mathcal{L}_{i-1}(\Theta_{i-1}, X_{i-1}) + \frac{1}{2}\|x(t_i) - f_o(x(t_{i-1}), u(t_{i-1}), \theta(t_{i-1}))\|_{Q_x^{-1}}^2
$$

$$
+ \frac{1}{2}\|\theta(t_i) - g_o(\theta(t_{i-1}))\|_{Q_\theta^{-1}}^2 \tag{21}
$$

$F_{x_{i-1}} = \frac{\partial f_o(x(t_{i-1}), u(t_{i-1}), \theta(t_{i-1}))}{\partial x(t_{i-1})}$

we define the matrix $L_x = [0_{d_x \times d_x}, \dots, 0_{d_x \times d_x}, I_{d_x \times d_x}]$ where $L$ is of dimensions $d_x \times ((i-1) \times d_x))$

by taking the gradient of equation (21) with respect to $X_i$ we obtain:

$$
\nabla\mathcal{L}_{i|i-1}^{x}(X_i, \Theta_i) = \begin{bmatrix} \nabla\mathcal{L}_{i-1}(X_{i-1}, \Theta_{i-1}) + L_x^T F_{x_{i-1}}^T Q_x^{-1}[x(t_i) - f_o(x(t_{i-1}), \theta(t_{i-1}))] \\ Q_x^{-1}[x(t_i) - f_o(x(t_{i-1})), \theta(t_{i-1})] \end{bmatrix} \tag{22}
$$

To minimize (21), we define the estimate $\hat{X}_{i|i-1}$ of $X_i$ to be the minimizer of (21). by setting $\nabla\mathcal{L}_{i|i-1}^{x}(X_i, \Theta_i)$ to zero we get the following:

$$
\hat{X}_{i|i-1} = \begin{bmatrix} \hat{X}_{i-1} \\ f_o(\hat{x}(t_{i-1}), \hat{\theta}(t_{i-1})) \end{bmatrix} \tag{23}
$$

with $\hat{x}(t_i|t_{i-1}) = f_o(\hat{x}(t_{i-1}), \hat{\theta}(t_{i-1}))$

then, we proceed to minimize the following equation with respect to $\Theta_i$:

$$
\mathcal{L}_{i|i-1}^{\theta}(\Theta_i, X_{i-1}) = \mathcal{L}_{i-1}(\Theta_{i-1}, X_{i-1}) + \frac{1}{2}\|\theta(t_i) - g_o(\theta(t_{i-1}))\|_{Q_\theta^{-1}}^2 \tag{24}
$$

$L_\theta = [0_{d_\theta \times d_\theta}, \dots, 0_{d_\theta \times d_\theta}, I_{d_\theta \times d_\theta}]$ where $L_\theta$ is of dimensions $d_\theta \times ((i-1) \times d_\theta))$

we take the gradient of equation (24) with respect to $\Theta_i$ we obtain:

$$
\nabla\mathcal{L}_{i|i-1}^{\theta}(\Theta_i, X_{i-1}) = \begin{bmatrix} \nabla\mathcal{L}_{i-1}^{\theta}(\Theta_i, X_{i-1}) - L_\theta^T G_{\theta_{i-1}}^T Q_\theta^{-1}[\theta(t_i) - g_o(\theta(t_{i-1}))] \\ Q_\theta^{-1}[\theta(t_i) - g_o(\theta(t_{i-1}))] \end{bmatrix} \tag{25}
$$

with $G_{\theta_{i-1}} = \frac{\partial g_o(\theta(t_{i-1}))}{\partial \theta(t_{i-1})}$

To minimize (24), we define the estimate $\hat{\Theta}_{i|i-1}$ of $\Theta_i$ to be the minimizer of (24). by setting $\nabla\mathcal{L}_{i|i-1}^{\theta}(\Theta_i, X_{i-1})$ to zero we get the following:

$$
\hat{\Theta}_{i|i-1} = \begin{bmatrix} \hat{\Theta}_{i-1} \\ g_o(\hat{\theta}(t_{i-1})) \end{bmatrix} \tag{26}
$$

the second step in the second order Newton method is to calculate the Hessian of $\mathcal{L}_{i|i-1}^{x}(\Theta_i, X_i)$:

$$
\nabla^2\mathcal{L}_{i|i-1}^{x}(\Theta_i, X_i) = \begin{bmatrix} \nabla^2\mathcal{L}_{i-1}^{x}(\Theta_{i-1}, X_{i-1}) + O(i) & L_x^T F_{x_{i-1}}^T Q_x^{-1} \\ Q_x^{-1} F_{x_{i-1}} L_x & Q_x^{-1} \end{bmatrix} \tag{27}
$$

where

$$
O(i) = -\frac{\partial^2 f(x(t_{i-1}), \theta(t_{i-1}))}{\partial^2 X_{i-1}} Q^{-1}[x(t_i) - f(x(t_{i-1}), \theta(t_i))] + L_x^T F_{x_{i-1}} Q_x^{-1} F_{x_{i-1}} L_x
$$

when $X_i = \hat{X}_{i|i-1}$ it follows that $O(i) = L_x^T F_{x_{i-1}}^T Q_x^{-1} F_{x_{i-1}} L_x$. using Lemma B.3 in Humpherys et al. (2012), the lower block $P_{x_i}^-$ of $\nabla^2\mathcal{L}_{i|i-1}^{x}(\Theta_i, X_i)$ is calculated as follows:

$$P_{x_i}^- = Q_x^{-1} + F_{x_{i-1}}^T P_{x_{i-1}} F_{x_{i-1}} \tag{28}$$

we continue to minimize

$$\mathcal{L}_i^x(X_i, \Theta_i) = \mathcal{L}_{i|i-1}^x(X_i, \Theta_i) + \frac{1}{2}\|y(t_i) - h(x(t_i))\|_{R_y^{-1}}^2 \tag{29}$$

we denote

$$\mathcal{H}_i = \left[0, 0, \dots, \frac{\partial h(x(t_i))}{\partial x(t_i)}\right]$$

. and by $H_i = \frac{\partial h(x(t_i))}{\partial x(t_i)}$ by taking the gradient of equation (29) we obtain:

$$\nabla \mathcal{L}_i^x(X_i, \Theta_i) = \nabla \mathcal{L}_{i|i-1}^x(X_i, \Theta_i) + \mathcal{H}_i R_y^{-1}(y(t_i) - h(x(t_i)))$$

The hessian of (29) becomes:

$$\nabla^2 \mathcal{L}_i^x(X_i, \Theta_i) = \nabla^2 \mathcal{L}_{i|i-1}^x(X_i, \Theta_i) + \frac{\partial^2 \mathcal{H}_i}{\partial^2 X_i} R_y^{-1}(y(t_i) - h(x(t_i))) + \mathcal{H}_i R_y^{-1} \mathcal{H}_i \tag{30}$$

setting $X_i = \hat{X}_{i|i-1} \implies \nabla \mathcal{L}_i^x(X_i, \Theta_i) = 0$ therefore:

$$\nabla \mathcal{L}_i^x(X_i, \Theta_i) = \begin{bmatrix} 0 \\ H_i R_y^{-1}(y_i - h_i(\hat{x}(t_i|t_{i-1}))) \end{bmatrix} \tag{31}$$

The hessian hence becomes:

$$\nabla^2 \mathcal{L}_i^x(X_i, \Theta_i) = \nabla^2 \mathcal{L}_{i|i-1}^x(X_i, \Theta_i) + \mathcal{H}_i R_y^{-1} \mathcal{H}_i \tag{32}$$

then according to the Newton method, we can update our estimate of $x_i$ as follows:

$$\hat{X}_i = \hat{X}_{i|i-1} - \left(\nabla^2 \mathcal{L}_i^x\right)^{-1} \nabla \mathcal{L}_i^x \tag{33}$$

let $P_x$ be the bottom right block of $\left(\nabla^2 \mathcal{L}_i^x\right)^{-1}$, therefore

$$\begin{aligned} P_{x_i} &= P_{x_i}^- + P_{x_i}^- H_i \left(R_y - H_i P_{x_i}^- H_i^T\right) H_i P_{x_i}^- \\ &= \left((P_{x_i}^-)^{-1} + H_i^T R_y^{-1} H_i\right)^{-1} \end{aligned} \tag{34}$$

by taking the bottom row of the newton equation (33) we get

$$\begin{aligned} \hat{x}(t_i) &= \hat{x}(t_i|t_{i-1}) - \left[\left(\nabla^2 \mathcal{L}_i^x(\Theta_i, X_i)\right)^{-1}\right]_{i,:} \nabla \mathcal{L}_i^x(\Theta_i, X_i) \\ &= \hat{x}(t_i|t_{i-1}) - P_{x_i} H_i R_y^{-1}\left(h(\hat{x}(t_i|t_{i-1})) - y(t_i)\right) \\ &= \hat{x}(t_i|t_{i-1}) - K_i^x\left(h(\hat{x}(t_i|t_{i-1})) - y(t_i)\right) \end{aligned} \tag{35}$$

where $row_i$ corresponds to the $i^{th}$ row of matrix $\left(\nabla^2 \mathcal{L}_i^x(\Theta_i, X_i)\right)^{-1}$ and

$$\begin{aligned} K_i^x &= P_{x_i} H_i R_y^{-1} \\ &= P_{x_i}^- H_i^T \left(H_i P_{x_i}^- H_i^T + R_y\right)^{-1} \end{aligned} \tag{36}$$

In a similar fashion we proceed to minimize

$$\begin{aligned} \mathcal{L}_i^\theta(\Theta_i, X_i) = \mathcal{L}_{i|i-1}^\theta(\Theta_i, X_{i-1}) &+ \frac{1}{2}\|x(t_i) - f_o(x(t_{i-1}), u(t_i), \theta(t_i))\|_{Q_x^{-1}}^2 \\ &+ \frac{1}{2}\|y(t_i) - h(x(t_i))\|_{R_y^{-1}}^2 \end{aligned} \tag{37}$$

$$\nabla \mathcal{L}_i^\theta(X_i, \Theta_i) = \begin{bmatrix} \nabla \mathcal{L}_{i|i-1}(X_{i-1}, \Theta_{i-1}) + L_\theta^T F_{\theta_{i-1}}^T Q_x^{-1} \left[x_i - f(x(t_{i-1}), \theta(t_{i-1}))\right] \\ Q_\theta^{-1}[\theta(t_i) - g_o(\theta(t_{i-1}))] \end{bmatrix} \tag{38}$$

where $F_{\theta_{i-1}} = \frac{\partial f_o(x(t_{i-1}), u(t_{i-1}), \theta(t_{i-1}))}{\partial \theta(t_{i-1})}$

at $\Theta_i = \hat{\Theta}_{i|i-1}$,

$$\nabla \mathcal{L}_i^\theta(X_i, \Theta_i) = \begin{bmatrix} L_\theta^T G_{\theta_{i-1}}^T Q_x^{-1} \left[x_i - f(x(t_{i-1}), \theta(t_{i-1}))\right] \\ 0 \end{bmatrix} \tag{39}$$

Similarly, the hessian of (29) is:

$$\nabla^2 \mathcal{L}_i^\theta(\Theta_i, X_i) = \begin{bmatrix} \nabla^2 \mathcal{L}_{i-1}^\theta(\Theta_{i-1}, X_{i-1}) + Z(i) & L_\theta^T F_{\theta_{i-1}}^T Q_\theta^{-1} \\ Q_\theta^{-1} F_{\theta_{i-1}} L_\theta & Q_\theta^{-1} \end{bmatrix} \tag{40}$$

where $Z(i) = \frac{\partial^2 f_o(x(t_{i-1}), u(t_{i-1}), \theta(t_{i-1}))}{\partial^2 \theta(t_{i-1})} Q_x^{-1} \left[x_i - f(x(t_{i-1}))\right] + L_\theta^T F_{\theta_{i-1}} Q_x^{-1} F_{\theta_{i-1}} L_\theta + L_\theta^T G_{\theta_{i-1}} Q_\theta^{-1} G_{\theta_{i-1}} L_\theta$ by ignoring second order terms we obtain: $Z(i) = L_\theta^T F_{\theta_{i-1}} Q_x^{-1} F_{\theta_{i-1}} L_\theta + L_\theta^T G_{\theta_{i-1}} Q_\theta^{-1} G_{\theta_{i-1}} L_\theta$

Then according to the newton second order method, we can update our estimate of $\Theta_i$ as follows:

$$\hat{\Theta}_i = \hat{\Theta}_{i|i-1} - \left(\nabla^2 \mathcal{L}_i^\theta\right)^{-1} \nabla \mathcal{L}_i^\theta \tag{41}$$

let $P_\theta$ be the bottom right block of $\left(\nabla^2 J_i^\theta\right)^{-1}$, therefore

$$\begin{aligned} P_{\theta_i} &= Q_\theta + G_{\theta_{i-1}} \left[ P_{\theta_{i-1}} - P_{\theta_{i-1}} F_{\theta_{i-1}}^T \left(Q_x + F_{\theta_{i-1}} P_{\theta_{i-1}} F_{\theta_{i-1}}^T\right) F_{\theta_{i-1}} P_{\theta_{i-1}} \right] G_{\theta_{i-1}} \\ &= Q_\theta + G_{\theta_{i-1}} P_\theta^- G_{\theta_{i-1}} \end{aligned} \tag{42}$$

where

$$P_{\theta_i}^- = P_{\theta_{i-1}} - P_{\theta_{i-1}} F_{\theta_{i-1}}^T \left(Q_x + F_{\theta_{i-1}} P_{\theta_{i-1}} F_{\theta_{i-1}}^T\right) F_{\theta_{i-1}} P_{\theta_{i-1}}$$

by taking the bottom of the newton equation (33) we get

$$\begin{aligned} \hat{\theta}(t_i) &= \hat{\theta}(t_i|t_{i-1}) - \left[\left(\nabla^2 \mathcal{L}_i^\theta(\Theta_i, X_{i-1})\right)^{-1}\right]_{i,:} \nabla \mathcal{L}_i^\theta(\Theta_i, X_{i-1}) \\ &= \hat{\theta}(t_i|t_{i-1}) + K_i^\theta \left(\hat{x}(t_i) - f(\hat{x}(t_i), \hat{\theta}(t_i))\right) \end{aligned} \tag{43}$$

with

$$K_i^\theta = G_{\theta_{i-1}} P_{\theta_i}^- F_{\theta_{i-1}}^T \tag{44}$$

# B   MODELS AND FURTHER EXPERIMENTS

## B.1   YEAST GLYCOLYSIS MODEL

Yeast glycolysis Model is a metabolic network that explains the process of breaking down glucose to extract energy in the cells. This model has been tackled by similar works in the field (Kaheman et al., 2020), (Mangan et al., 2016), and (Schmidt et al., 2011). It has seven states: $x = \begin{bmatrix} x_1 & x_2 & x_3 & x_4 & x_5 & x_6 & x_7 \end{bmatrix}^T$, and ODEs for these states are given from Eq (45), (Mangan et al., 2016).

$$\dot{x}_1 = c_1 + \frac{c_2 x_1 x_6}{1 + c_3 x_6^4},$$

$$\dot{x}_2 = \frac{d_1 x_1 x_6}{1 + d_2 x_6^4} + d_3 x_2 - d_4 x_2 x_7,$$

$$\dot{x}_3 = e_1 x_2 + e_2 x_3 + e_3 x_2 x_7 + e_4 x_3 x_6,$$

$$\dot{x}_4 = f_1 x_3 + f_2 x_4 + f_3 x_5 + f_4 x_3 x_6 + f_5 x_4 x_7, \qquad (45)$$

$$\dot{x}_5 = g_1 x_4 + g_2 x_5,$$

$$\dot{x}_6 = h_3 x_3 + h_5 x_6 + h_4 x_3 x_6 + \frac{h_1 x_1 x_6}{1 + h_2 x_6^4},$$

$$\dot{x}_7 = j_1 x_2 + j_2 x_2 x_7 + j_3 x_4 x_7$$

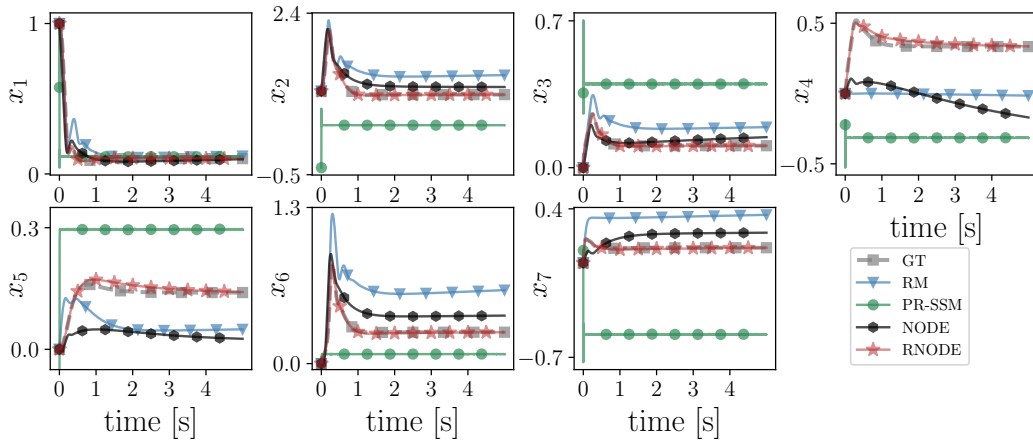

Figure 6: Learned state trajectories of yeast glycolysis model after training with the RM, PR-SSM, NODE, and RNODE methods. Results are compared to ground truth ODE system trajectory labeled as GT. We showed that the proposed approach (RNODE) is capable of discerning the true trajectory for the unobserved state $x_4$.

In this scenario, we assumed the dynamics of state $\dot{x}_4$ in equation 45 to be unknown and $x_4$ unobserved. We used the proposed RNODE and competing algorithms to learn NN-based dynamics. Specifically, we replaced $\dot{x}_4$ with a two-layer neural network with $tanh$ activation function on each layer. We generate a dataset with 5000 time steps, $dt = 10^{-3}s$ and by collecting measurements $\mathcal{D} \triangleq \{y(t_0), \ldots, y(t_{N-1})\}$. For the RNODE the hyper-parameters were set to $P_{x_0} = 10^{-2} I_{d_x}, P_{\theta_0} = 10^2 I_{d_\theta} \ R_y = 10^{-10} I_{d_y}, Q_x = 10^{-5} I_{d_x}$ and $Q_\theta = 10^{-2} I_{d_\theta}$. At the beginning of each epoch, we solve problem (58) of the Appendix C to get the initial condition. Final optimal parameters $\hat{\theta}(t_N)$ and initial condition $\hat{x}(t_0)$ are saved and collected at the end of training.

We assess the performance of RNODE by setting the model parameters to $\hat{\theta}(t_N)$ and perform integration following the model described in equation 3 starting from initial condition $\hat{x}(t_0)$. In Figure 6, we demonstrate the ability of the proposed RNODE to learn the underlying dynamics of the partially observed system compared to RM, PR-SSM and NODE methods. Table 1, shows that RNODE clearly outperforms the competing algorithms with nRMSE 99.3% , 99.1% and 90.4% smaller than the nRMSEs obtained by PR-SMM, RM, and NODE respectively. Analyzing the evolution of the latent states depicted in Figure 6, we notice that RNODE provides state trajectories that match the ground truth (GT). PR-SSM fails to capture the dynamics of the system, while RM and NODE presents acceptable trajectories of most of the states except for the unobserved dynamics of $\dot{x}_4$, and the observed dynamics of $\dot{x}_5$.

Moreover, both RM and NODE state trajectories are much more inaccurate than the one provided by RNODE. This shows the challenging nature of the problem and the proposed approach's efficiency in learning challenging nonlinear dynamics using estimates of the unobserved states, which is unavailable to the other methods.

## B.2 HODGKIN-HUXLEY NEURON MODEL

For the HH model, we refer to the (Hodgkin & Huxley, 1952) and use the following ODE system. The ODE system has four states: $V_m$, $n_{gate}$, $m_{gate}$, and $h_{gate}$. $I_e$ is the external current input, which is set to 10 if the neuron is firing, and 0 otherwise. For all models, we simulate the dynamics of the HH model with a time step of 0.01 ms and integrate using Euler integration.

$$\dot{V}_m = I_e - 36n_{gate}^4(V_m + 77) - 120m_{gate}^3 h_{gate}(V_m - 50) - 0.3(V_m + 54.4) \tag{46}$$

$$\dot{n}_{gate} = 0.01(V_m + 55)\left[1 - \exp\left(-\frac{V_m + 55}{10}\right)\right]^{-1}(1 - n_{gate}) - 0.125\exp\left(-\frac{V_m + 65}{80}\right)n_{gate} \tag{47}$$

$$\dot{m}_{gate} = 0.1(V_m + 40)\left[1 - \exp\left(-\frac{V_m + 40}{10}\right)\right]^{-1}(1 - m_{gate}) - 4\exp\left(-\frac{V_m + 65}{18}\right)m_{gate} \tag{48}$$

$$\dot{h}_{gate} = 0.07\exp\left(-\frac{V_m + 65}{20}\right)(1 - h_{gate}) - \left[1 + \exp\left(-\frac{V_m + 35}{10}\right)\right]^{-1}h_{gate} \tag{49}$$

## B.3 RETINAL CIRCULATION MODEL

The retinal circulation model describes the internal pressures of five compartments in the retina (Guidoboni et al., 2014). The model has four states: $P_1$, $P_2$, $P_4$, and $P_5$. The relation between these states is summarized in Eqs. (50)-(53). In our experiments, we don't measure $P_5$ and set $y = (P_1, P_2, P_4)$ then train RM, PR-SSM, and RNODE to approximate the ODE trajectories. In Fig. 7, we visualize the state trajectories for all states and demonstrate that RNODE outperforms PR-SSM and RM at estimating state trajectories, RNODE model successfully captures the dynamics of the unmeasured $P_5$ state.

$$\dot{P}_1 = \frac{P_{in} - P_1}{C_1(R_{in} + R_{1a})} - \frac{P_1 - P_2}{C_1(R_{1b} + R_{1c} + R_{1d} + R_{2a})} \tag{50}$$

$$\dot{P}_2 = \frac{P_1 - P_2}{C_2(R_{1b} + R_{1c} + R_{1d} + R_{2a})} - \frac{P_2 - P_4}{C_2(R_{2b} + R_{3a} + R_{3b} + R_{4a})} \tag{51}$$

$$\dot{P}_4 = \frac{P_2 - P_4}{C_4(R_{2b} + R_{3a} + R_{3b} + R_{4a})} - \frac{P_4 - P_5}{C_4(R_{4b} + R_{5a} + R_{5b} + R_{5c})} \tag{52}$$

$$\dot{P}_5 = \frac{P_4 - P_5}{C_5(R_{4b} + R_{5a} + R_{5b} + R_{5c})} - \frac{P_5 - P_{out}}{C_5(R_{5d} + R_{out})} \tag{53}$$

$R_{in}, R_{1a}, R_{1b}, R_{2a}, R_{2b}, R_{3a}, R_{3b}, R_{5c}, R_{5d}$, and $R_{out}$ are fixed resistances. $R_{4a}, R_{4b}, R_{5a}$, and $R_{5b}$ depend on states. $C_{1-5}$ are the constant capacitance values. $P_{in}$ is time-varying input, and $P_{out}$ is constant output which is set to 14.

The numerical results for the retinal circulation model experiments are summarized in Table 1 and visual results are presented in Figure 7. Both results show that RNODE clearly outperforms the competing algorithms with nRMSE value that is 94.3% and 28.8% smaller than the nRMSEs obtained by PR-SMM and RM, respectively. Analyzing the evolution of the latent states depicted in Figure 7, we notice that RNODE provides state trajectories that match the ground truth (GT) more closely when compared with PR-SSM and RM. Similar to PR-SSM results for the HH, HO, and EMPS models, PR-SSM again presents very poor state trajectories indicating that the model was not capable of learning the underlying ODE function accurately. nRMSE value for the RM is comparable to RNODE, however, the lower right panel of the Fig. 7 pinpoints that RNODE excels in learning the unobserved state $P_5$.

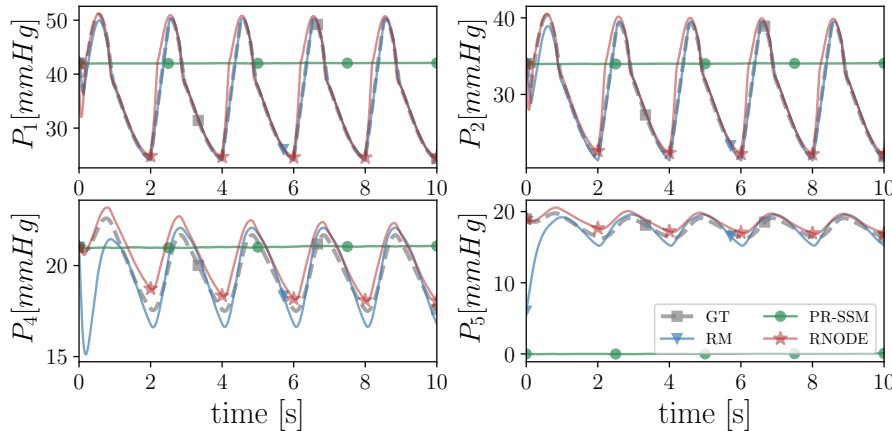

Figure 7: Estimated state trajectories of Retinal circulation model after training RM, PR-SSM, and RNODE methods. Results are compared to ground truth ODE system trajectory labeled as GT. We showed that the proposed approach (RNODE) is capable of discerning the true trajectory for the unobserved state $P_5$.

### B.4 HARMONIC OSCILLATOR

The harmonic oscillator has two states, representing position and velocity, where $z$ is the position and $\dot{z}$ is the velocity: $x = \begin{bmatrix} z & \dot{z} \end{bmatrix}^T$. $u$ is the input vector which is set to zero for a free harmonic oscillator: $u = [0]$. $\omega$ is the unknown angular frequency. State equations can be written in matrix form as follows:

$$\dot{x} = \begin{bmatrix} 0 & 1 \\ -\omega^2 & 0 \end{bmatrix} x + \begin{bmatrix} 0 \\ 1 \end{bmatrix} u$$

Here, the matrix $\begin{bmatrix} 0 & 1 \\ -\omega^2 & 0 \end{bmatrix}$ is the state transition matrix and it represents the system's dynamics. Throughout the experiments, we only observed position state $x_1$, hence, our output equation is: $y = \begin{bmatrix} 1 & 0 \end{bmatrix} x$. We simulated the ODE system using Euler integration and we used a time step of 1 ms.

The numerical results for the HO experiments are summarized in Table 1 while the corresponding visual results can be found in Figure 8. Both results clearly demonstrate RNODE's superior performance against PR-SSM and NODE, and Table 1 shows a modest advantage over RM. RNODE achieved notably smaller nRMSE. Indeed, the nRMSE achieved using RNODE is 99%, 78% and 98% smaller than the nRMSEs obtained by PR-SMM, RM and NODE respectively. Analyzing the evolution of the states in Figure 8, we notice that PR-SSM failed to learn the underlying ODE function accurately and that NODE failed to learn the period of the signal.

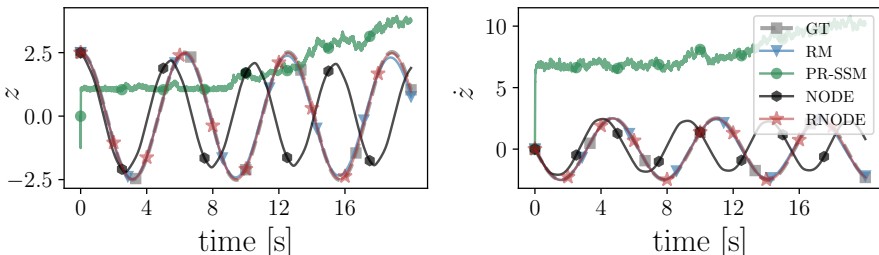

Figure 8: Estimated state trajectories of harmonic oscillator after training RM, PR-SSM, NODE, and RNODE methods. Results are compared to ground truth ODE system trajectory labeled as GT. We showed that the proposed approach (RNODE) is capable of discerning the true trajectory for the unobserved state $\dot{z}$.

Figure 9 depicts the learned ODE vector field (left) and true vector field (right). We can observe that RNODE was capable to learn resonably well the true ODE function.

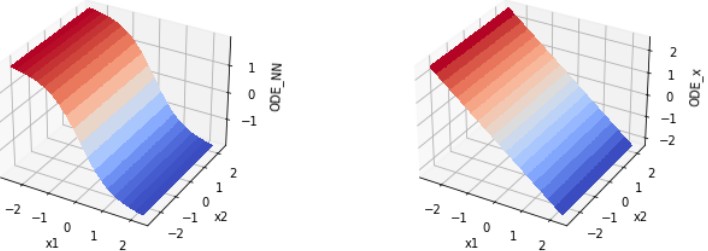

Figure 9: Learned ODE_NN (left) and true ODE_x (right) vector fields of the Harmonic Oscillator model after training with the RNODE method. We showed that the proposed approach (RNODE) is capable of learning the true vector field.

### B.5 CART-POLE SYSTEM

The cart-pole system is a classic problem in control theory and it models the movement of a cart along an axis, and this cart has a pole attached to it and this pole can pivot freely. States of the ODE system can be defined as: $x = \begin{bmatrix} z & \dot{z} & \phi & \dot{\phi} \end{bmatrix}^T$. We linearize the system at $x = \begin{bmatrix} z & \dot{z} & \phi & \dot{\phi} \end{bmatrix}^T = \begin{bmatrix} 0 & 0 & 0 & 0 \end{bmatrix}^T$ and use LQR controller (Prasad et al., 2011) to calculate the input $u$ that swings the pendulum and balances it in the inverted position in the middle of the track.

$$\dot{z} = \dot{z} \tag{54}$$

$$\ddot{z} = \frac{-m \cdot l \cdot \sin(\phi) \cdot \dot{\phi} + u + m \cdot g \cdot \cos(\phi) \cdot \sin(\phi)}{M + m - m \cdot \cos(\phi)^2} \tag{55}$$

$$\dot{\phi} = \dot{\phi} \tag{56}$$

$$\ddot{\phi} = \frac{-m \cdot l \cdot \cos(\phi) \cdot \sin(\phi) \cdot \dot{\phi}^2 + u \cdot \cos(\phi) + m \cdot g \cdot \sin(\phi) + M \cdot g \cdot \sin(\phi)}{l \cdot (M + m - m \cdot \cos(\phi)^2)} \tag{57}$$

In these matrices: $M$ is the mass of the cart, $m$ is the mass of the pole, $l$ is the length from the cart's center to the pole's center of mass, $l_c$ is the length from the cart's center to the pivot point, and $g$ is the acceleration due to gravity.

## C INITIAL CONDITION RECONSTRUCTION DURING TRAINING

Given a model $\theta$ and a dataset $\mathcal{D} \triangleq \{u(t_0), y(t_0), \ldots, u(t_{N-1}), y(t_{N-1})\}$ , training the model requires determining an appropriate initial state $x(t_0)$ at the beginning of each epoch. A way to get $x(t_0)$ is to solve the following state-reconstruction problem:

$$\min_{x(t_0|t_{-1})} \quad \|y(t_0) - h(\hat{x}(t_0))\|^2_{R_y^{-1}} \tag{58}$$

In this case, Problem 58 can provide a suitable value for $\hat{x}(t_0|t_{-1})$ for the new epoch based on the last vector $\theta(t_{N-1})$ learned, that is used in 58 and as the initial condition $\theta(t_0|t_{-1})$ for the new epoch. We remark that when RNODE is run on $N_e$ epochs and $P_{x_0}^-$ is set equal to the value $P_{x_N}^-$ from the previous epoch. $x(t_0)$ is computed next according to equation (16).

## D COMPLEXITY ANALYSIS

To calculate the complexity of RNODE's training procedure, first, we need to calculate the complexity of each Jacobian. Since we are using automatic differentiation to calculate them, the complexity

of getting $F_{x_{i-1}}, G_{\theta_{i-1}}$, and $F_{\theta_{i-1}}$ is $\mathcal{O}(d_x^2), \mathcal{O}(d_\theta^2)$ and $\mathcal{O}(d_\theta d_x)$, respectively. However, the complexity of $G_{\theta_{i-1}}$ could be removed if $g(\theta(t_i)) = 0$ since $G_{\theta_{i-1}} = I$ in that case. Note that this is the case in our experiments.

Assuming $n_x \approx n_y$, the complexity of calculating $P_{\theta_i}^-, P_{x_i}^-, P_{x_i}$ and $P_{\theta_i}$ are $\mathcal{O}(2d_\theta^2 d_x + d_x + (2d_x^2 + 1)d_\theta), \mathcal{O}(2d_x^3 + d_x), \mathcal{O}(4d_x^3 + 2d_x)$ and $\mathcal{O}(d_\theta)$, respectively.

Moreover, the complexity of computing $\hat{x}(t_i)$ is $\mathcal{O}(2d_x^3 + d_x^2)$, and the complexity of $\hat{\theta}(t_i)$ is $\mathcal{O}(d_\theta(d_x + 1))$.

Since the Jacobians are differentiated once, and then evaluated at each time step, the complexity of one epoch performed over $N$ samples becomes:

$$\begin{aligned}
\mathcal{O}(d_x d_\theta) &+ \mathcal{O}(d_\theta^2) + \mathcal{O}(d_x^2) + N[\mathcal{O}(2d_\theta^2 d_x + (2d_x^2 + 1)d_\theta) + \mathcal{O}(2d_x^3 + d_x) + \mathcal{O}(4d_x^3 + 2d_x) \\
&+ \mathcal{O}(d_\theta) + \mathcal{O}(2d_x^3 + d_x^2) + \mathcal{O}(d_\theta(d_x + 1))]
\end{aligned} \tag{59}$$

which simplifies to

$$\mathcal{O}(N(d_x^3 + d_\theta^2 d_x + d_x^2 d_\theta)). \tag{60}$$

Finally, assuming $d_\theta \gg d_x$, the total cost of each training epoch can be simplified as:

$$\mathcal{O}(N(2d_\theta^2 d_x + 2d_\theta d_x^2)). \tag{61}$$

It is clear that the main source of complexity would be $d_\theta$. Thus, RNODE may scale badly to very higher-dimensional neural network architectures. However, fast approximations of Newton's method exist, as pointed out by the reviewer, such as Shampoo. Although merging Shampoo with our proposed approach could reduce the computational burden, we will analyze this hypothesis in future works.

**Test-time complexity:**   Although in our experiments we just integrated the learned mean vector field, RNODE can be employed in different ways depending on the availability of data (data assimilation). RNODE can be also used in an online learning paradigm where learning and estimation are continuously performed. Thus, the computational complexity per time-step in these scenarios becomes $\mathcal{O}(d_\theta), \mathcal{O}(d_x^3)$, and $\mathcal{O}(d_x^3 + d_\theta^2 d_x + d_x^2 d_\theta)$, for mean vector field integration, data assimilation, and continuously assimilation and adaptation, respectively.

