# OpenReview forum: "RECURSIVE NEURAL ORDINARY DIFFERENTIAL EQUATIONS FOR PARTIALLY OBSERVED SYSTEM"
_ICLR.cc/2024/Conference — ICLR 2024 Conference Withdrawn Submission_

### Official Review · Reviewer_QsPz · 2023-10-30

**Soundness:** 3 good
**Presentation:** 2 fair
**Contribution:** 2 fair
**Rating:** 3
**Confidence:** 3

**Summary:**

This paper proposes a Newtonian optimisation for ODEs where some state dimensions are not measured, and where some state dimension differential are not known. The method can fit simple partially observed ODE systems exactly.

**Strengths:**

- The performance is top notch
- The method is generally convincingly presented

**Weaknesses:**

- The presentation is a bit confusing at times, and open problem and contributions are not clear. The method contains Hessians and parameters processes that are eventually not used, convoluting the model. Some notation and math is confusing. Main claim is not substantiated (vanishing adjoints).
- There is no comparison to baselines, which would be standard neural ODE loss, or towards multiple shooting layers (Massaroli et al). Comparison to some SINDy method would be good as well. There is no running time analysis
- There is no real-world experiment: all demonstrations are simple toy cases with no unknown parts. The paper should include at least one case where the underlying process is not fully known or too messy or difficult (eg. motion capture, biochemistry, etc).

**Questions:**

Minor comments

- It’s odd to call the dx as “diffusion”: it’s not diffusion, but a differential, or just an ODE. I wonder what the x and theta represents in this work.
- I’m not sure I get where the def 3.1. is coming from, or why its true. There is no citation, or proof. Naively of course we can’t distinguish states that emit the same observation, but we could also have a situation where one state is going up, one is going down, and they cross at one point. Even if they are identical at one point, does not mean that we can’t distinguish them based on the surrounding information, or from the slopes. Similarly, one could have a cyclical system that visits same state many times (eg. VDP). There again having same output from two states is fine.
- Suddenly in 3.3. the theta is network parameters. I’m a bit confused by this. I thought the purpose of the parameter process was that the system is non-stationary over time. Is this still accurate?
- I don’t understand how an un-measurable state makes adjoints zero. Surely if we have a partially observed system, the neural ODE will happily fit the trajectory with the known states as well as it can. For instance, if we have a 3D lorentz attractor where we only know about the first 2 dimensions, we can easily still fit an observed 2D trajectory without vanishing if we include evolving parameters or time-dependent system. The paper needs to demonstrate its claims by proofs or citations, and preferably also give a proof-of-concept toy illustration of the effect as well.
- The notation in eq 5 is misleading: why do we parameterise by u(t_i-1) but not by u(t)? Surely we need to know the u, x and theta for all arbitrary timepoints between t_i and t_i-1. The ODEsolve only refers to x, while you also need to solve for theta
- I’m confused by eq 8. I’m not sure what is your model here. I would assume that you have a neural network that represents dtheta and dx. But neural network parameters are nowhere to be seen. Is \theta neural network parameters?
- The presentation of the piecewise optimisation is quite confusing. I don’t get where the Hessians are coming from, or what’s their point in the first place. Apparently h doesn’t have parameters, so it has to be known (but if states are not fully observed, how can you know h?). I don’t get the joint vanishing statements. I can’t follow the eqs 10 and 11. It seems that you just set the x and theta to the ODE solutions. Ok, so you are just an ODE solve here. Why do we then need the eqs 10 and 11 at all?
- When you say that you optimise for x(t_i), what does that mean? Do optimise for the realisation of x(t_i), or its parameters? Btw. x(t_i) is not dependent on parameters notationally, which is very confusing. One would expect instead x(t_i | x_0, params).
- I don’t see how eqs 14 and 16 are any different from standard neural ODE loss of eq 3. To me this is a reordering of the original simple loss into a more complicated version, which still looks like the same thing. The eqs 17-19 seem to be just gradient updates with Hessians. Ok, but can’t we directly use a second-order optimiser for the standard neural ODE loss? What is the point of all of this? How does this help us reveal the unknown dimensions of x?
- What does “known” mean in thm 4.1? Is it the value of the true underlying system?
- If all experiments have g=0, why include it in the model at all?
- The experiments need to compare to standard neural ODE loss as reference, with both known or unknown initial value.
- The paper is missing citation to DMSL (Massaroli et al 2021). This is also a Newtonian optimiser, so one should directly compare against it, and also discuss how it differs.
- In experiments the Hessians (eg. Q) are just diagonal. What was the point of this if you don’t effectively use them?

---

> ### Author Response · Authors · 2023-11-22
> **Reply to the Weaknesses**
>
> We thank the reviewer for the detailed comments which definitely improved the clarity and quality of the revised manuscript.
> Below, we reply to all Weaknesses provided by the reviewer point by point.
> When referring to equations we refer to the numbers in the original manuscript unless explicitly stated.
>
>
> - **Weakness 1:**
>
> We agree that the presentation, especially the positioning of the method, was a bit confusing in the original manuscript. Besides, we would like to highlight that the proposed approach is not based on the adjoint method, and that giving too much emphasis to NODE and its possible problems may have added to the confusion the reviewer experienced.
> In the case of the proposed RNODE approach, we leveraged an alternating optimization to avoid vanishing gradients that would be caused by our employed methodology. Again this point is not necessarily related to the standard NODE.
>
> In fact, after the reviewer's comments, we noticed that in the case of standard NODE, the initial condition of the adjoints does indeed vanish, but its integral does not.
> For this reason, and to improve the clarity of the paper regarding our contributions, we removed Section 3.3 of the original manuscript and revised our discussions regarding vanishing gradients in the manuscript. Furthermore, we included results with NODE in all experiments.
>
> It becomes evident that NODE leads to poor performance for the more challenging scenarios since it was not designed to deal with unobserved latent states.
> We thank the reviewer for their critical review which certainly improved the clarity of our contributions and positioning of the RNODE approach.
>
> The open problem is stated in the abstract, in the second paragraph of the introduction and related work section. The contribution of the proposed method is also discussed at the end of the introduction, and in the related work sections.
> To emphasize both the open problem and contribution we modified to third paragraph of the introduction to improve the clarity of the problem and contributions. We also introduced a paragraph "The partial observation problem" at the end of Section 3.1 to address this issue.
>
> Regarding the usage of Hessians, they are always used. In Appendix A, we provide the full derivation of Theorem~1 where the connection of the different quantities P’s, Q’s, R’s, etc, and the Hessians are clearly shown.
>
> Although in our experiments we consider a small-variance random walk, mainly for numerical reasons, for the parameters process the formulation is general and other processes could be used. For this reason, we kept the general parameter process formulation presented in Section 3.1.
>
>
> - **Weakness 2:**
>
> We would like to point out that we did not compare it with vanilla ODE in our original manuscript because existing NODE is not demonstrated to learn with partially observed state constraint. Nevertheless, we agree with the reviewer that standard neural ODE comparison with RNODE would be valuable and provide more context, that’s why we modified our experimental results section to include benchmarks of RNODE against the vanilla NODE approach.
>
> Regarding SINDy's strategies, to our knowledge, they are not designed to deal with unobserved states. Furthermore, SINDy does not learn global models. Instead, SINDy learns a local model $\Xi_k$ at each time step $k$. This is why we did not compare with it as we are learning a global model.
>
> Concerning the run-time analysis, we included the complexity of the proposed algorithm in Appendix D of the revised manuscript. We refer to it at the end of Section 4.1.
>
> -  **Weakness 3:**
>
> Thank you for your suggestion, however, we would like to clarify that we did include a real-world experiment which is the Electro-Mechanical Positioning System. Moreover, even though we believe the Hodgkin-Huxley model is fairly complex, we added a complex biochemistry example, namely, \textit{yeast glycolysis} to Appendix B.

---

> ### Author Response · Authors · 2023-11-22
> **Reply to the Questions 1-7**
>
> Below we reply to the Questions point by point.
>
> - We agree with the reviewer. We refer to $dx$ and $d\theta$ as stochastic differential equations in the revised manuscript as defined in [5]. We also modified the definitions in Section 3.1, and also the generative model in Section 3.4 of the original manuscript to cope with the stochasticity of the approach. We apologize for the confusion.
> Regarding the second part of the question,
> $x$ represents the dynamical system states,  which we want to learn parts of its dynamics using neural networks parameterized by $\theta$.
> In the more general scenario of non-stationary processes, $\theta$ may have non-zero dynamics to cope with non-stationary scenarios such as [1], to speed up the learning procedure, or to avoid numerical issues.
>
> - We understand the reviewer's point. Nevertheless, we opted for this simplistic and at the same time, very restrictive condition, since we require some type of distinguishability.
> We would like to point out that states following Definition 3.1 lead to unique observations. This is necessary for recovering unique states from the observations. Without this condition, there is no guarantee that states can be uniquely recovered from the observations. A simple example is $y = x^2$ where y represents the observations and $x$ is the latent variable. It is clear that in this scenario one cannot uniquely recover $x$ from $y$. We agree that such a condition is sufficient but may not be necessary. However, determining the necessary conditions for state estimation in dynamical models with neural networks is a very challenging open problem and is outside the scope of this paper.
>
> - The work focuses on learning unknown dynamics from data where the state related to the unknown dynamics is unobserved. Since the dynamics is unknown, we assume parametric models such as NNs.
> We clarified this point in the third paragraph of Section 1.
> Regarding non-stationarity, although the RNODE approach could indeed be used to deal with such systems, in this paper we learn a constant dynamical model as shown in our experiments.
>
> - The reviewer is correct. We revised this point and noticed that although the initial condition of the adjoint vanishes, this is not necessarily the case for it's integral as we pointed out in the reply for Weakness 1.
>
> - To generate $x(t_i)$ in Equation 5 (Equation 4 in the revised manuscript), we need $x(t_{i-1})$ and $u(t_{i-1})$ and hence  $x_{t_i}=f_o(x(t_{i-1}),u(t_{i-1}))$. Note that this parametrization of input variables is ubiquitous in the dynamical system literature [2].
> Please, note the solution for $\theta$ in Equation 6 (Equation 5 in the revised manuscript).
>
> - Note that in the text before Equation 1 we define $\theta$ as the system's parameters. In the scenario where part of the dynamics are modeled by NNs, then $\theta$ also refers to the parameters of the NNs. We refer to $x$ as dynamical system states which are modeled by $f$. In our work, we tested the proposed approach using hybrid models where partial knowledge of $f$ was assumed, and where the dynamics of non-observed states were considered unknown and modeled as a neural network parameterized by $\theta$. We emphasized this point in the third paragraph of the Introduction in the revised manuscript.
>
> - The Hessians come from the application of Newton's method as shown in Appendix A. In the Appendix we show how the Hessians can be written as functions of other quantities $P$'s, $Q$'s, etc, leading to the update expressions shown in Theorem 4.1.
> We assume the observation function $h$ to be known for the known states.
> For instance, in a 2D system, if $h=I$, $y=[x_1,x_2]$, if we observe $x_1$, then $H=[1 \quad 0]$. This copes with many systems and applications in control, physiological systems modeling, etc, where the measurement functions for known states are known.
>
> Considering the vanishing adjoints, see the reply to Weakness 1.
>
> Regarding Equations 10 and 11 in the original manuscript, the reviewer's understanding is inaccurate since we do not just set $x$ and $\theta$ to the ODE solution.
> We split the problem in Equation 9 (in the original manuscript) into four sub-problems, see, Equations 10, 11, 13, and 15 of the original manuscript, to obtain alternating optimizations for $x$ and $\theta$ for prediction (Eqs. 10 and 11) and update steps (Eqs. 13 and 15).
> The benefit of this approach is that we can estimate $\theta$ in (15) using the solution for $x$ found in (13) which contains the estimated unobserved state of the hidden dynamics. Finally, $x(t_i)$ and $\theta(t_i)$ are updated at each time step according to (17) and (18) in the original paper.
> We modified Section 4.1 to clarify these points.

---

> > ### Author Response · Authors · 2023-11-22
> > **Reply to Questions 8-14**
> >
> > - When minimizing the cost with respect to $x$ means that one searches for the $\hat{x}$ that leads to the smaller value of the cost function.
> > This is standard in sequential state-estimation theory [2,4], the state estimation procedure can be split into two steps: a prediction step and an update step, leading to problems to be optimized with respect to the states. Note that in the proposed approach we have to optimize with respect to both states and parameters which we do in an alternating manner leading to the benefits discussed in the previous reply.
> >
> > - We understand the reviewer's confusion. First, note that eqs. (14) and (16) of the original manuscript are the solutions obtained with the RNODE approach that we detailed in the appendix and the previous reply. The original cost function solved by RNODE is presented in (8) and differs from the standard NODE loss $\mathcal{L}^\mathrm{NODE}=\sum_i\|y(t_i) - h(\hat{x}(t_i))\|^2$.
> > We further split the cost function, leading to a 4-step alternate optimization strategy (see, eqs. (10), (11), (13) and (15) of the original paper ) that allows the states to approximate the observations. We described this process in the previous reply. This discussion appears in the paragraph after Theorem 4.1.
> >
> > - We mean given. We agree that this terminology was misused. We fixed this in the revised manuscript.
> >
> > - That is true. We added $g$ since different dynamics could be used for $\theta$. In general, $g$ can take different forms to address different stationary and non-stationary processes such as in [1]. For this reason, we opted to keep $g$ in the model.
> >
> > - We thank the reviewer for the suggestion, we added the comparison with NODE with standard loss for each experiment. Note, however, that the NODE approach [3] presents no approach for estimating initial conditions and will categorically fail in this scenario. We compared our results with known initial conditions for the NODE.
> >
> > - Thank you for pointing that out, we added it to our related work section, Nevertheless, we didn't have time to compare against it experimentally, since it also does not deal with learning with partial observations.
> >
> > - We would like to point out that $Q$ is not the Hessian, $Q$ and $R$ are positive definite matrices in the cost function. The Hessians are defined in eqs. (28),(31),(33),(41) of the original manuscript in the appendix. Regarding $Q$, it could indeed be a full matrix if knowledge regarding the correlation among states, or correlation among parameters is available. In the filtering literature, there are many strategies to estimate such quantities, including the employment of the Expectation Maximization algorithm. This, however, is outside of the scope of the present contribution.
> >
> >
> > [1] Schmidt, J., Krämer, N. and Hennig, P., 2021. A probabilistic state space model for joint inference from differential equations and data. Advances in Neural Information Processing Systems, 34, pp.12374-12385.
> >
> > [2] Sarkka, Simo, and Lennart Svensson. Bayesian filtering and smoothing. Vol. 17. Cambridge university press, 2023.
> >
> > [3] Chen, R.T., Rubanova, Y., Bettencourt, J. and Duvenaud, D.K., 2018. Neural ordinary differential equations. Advances in neural information processing systems, 31.
> >
> > [4] Humpherys, J., Redd, P. and West, J., 2012. A fresh look at the Kalman filter. SIAM review, 54(4), pp.801-823.
> >
> > [5] Øksendal, B. and Øksendal, B., 2003. Stochastic differential equations (pp. 65-84). Springer Berlin Heidelberg.

---

> ### Comment · Reviewer_gkPz · 2023-11-22
>
> **On the reply on weakness 2:**
>
> >Regarding SINDy's strategies, to our knowledge, they are not designed to deal with unobserved states. Furthermore, SINDy does not learn global models. Instead, SINDy learns a local model $\Xi_k$ at each time step $k$. This is why we did not compare with it as we are learning a global model.
>
> SINDy is supposed to be learning a global model of the dynamics, not local ones. On your second point, it has been applied in conjunction with delay embedding to identify dynamical systems from partial observations [1].
>
>
>
>
> [1] Bakarji, J., Champion, K., Kutz, J. N., & Brunton, S. L. (2022). Discovering governing equations from partial measurements with deep delay autoencoders. arXiv preprint arXiv:2201.05136.

---

> > ### Author Response · Authors · 2023-11-22
> > **Reply to the Reviewer comment regarding SINDy**
> >
> > We thank the reviewer for pointing out this reference (which we were unaware of). We will include it in the related work section of the revised manuscript.  Unfortunately, we don’t have time to include this method in our experiments in this revision round. We apologize for the confusion and when looking at the original SINDy publication. Nevertheless, we included experiments with NODE as suggested by the reviewer.

---

> ### Author Response · Authors · 2023-11-23
> **Revised manuscript submitted**
>
> Dear reviewer,
>
> we submitted the revised manuscript. We believe to have addressed most of the reviewer's points and hope the revised manuscript matches your expectations. Thank you for your valuable input.

---

### Official Review · Reviewer_gkPz · 2023-11-01

**Soundness:** 2 fair
**Presentation:** 3 good
**Contribution:** 3 good
**Rating:** 5
**Confidence:** 3

**Summary:**

The authors propose a  framework for learning dynamics of partially observed ODE systems from observations based on the Neural ODEs (NODEs) formalism. They exploit the relationship between recursive-state space estimation procedures and Newton’s method to establish a sequential framework to learn latent states and model parameters.

To overcome the infeasibility of the optimisation of latent parameters $\theta$ and latent states $x$ when considering an optimisation cost that considers all observations (time steps) in one run, they employ a sequential/online optimisation procedure, where the optimisation cost at each time step is expressed recursively, in terms of the optimisation cost of the previous time step.

Assuming distinguishable latent states, they decouple the sequential optimisation for latent states and parameters, and propose alternating Newton updates to estimate the respective parameters/states at each time step.
They demonstrate the performance of the proposed framework on a battery of model systems, where they assume that part of the model dynamics is unknown.

The main contribution of the paper is the sequential formulation of the optimisation of latent parameters and states outlined in detail in Theorem 1 in the Supplement/Appendix.

**Strengths:**

- The paper proposes a sequential optimisation for latent states and variables of partially unknown ODE systems that is new to my knowledge.
- They demonstrate the accuracy of their method on several model systems.

**Weaknesses:**

- In the presentation of the modeling assumptions of their framework (Section 3.1) the authors mention that they assume a latent diffusion model and Markovian evolution of the model parameters ($\theta(t)$), however in the equations they set out  in the same section and in their numerical experiments there is no stochastic term considered. IN Section 3.4 (Eq.7) they indeed consider a deterministic system with  and I assume they mention Markovian dynamics to indicate that each state depends only on the state of the previous time step, but to my knowledge Markovianity also implies stochasticity which is not apparent here. But I am open to be corrected.
This is quite confusing/misleading to me, unless the authors clarify their assumptions.

- The framework requires the latent states ($x$) of the system to be distinguishable, uniquely identifiable from the observations given a certain control input. However, I find this condition quite limiting for realistic applications, and I wonder whether this is a hard requirement, or whether the method could still perform quite well in settings with partially indistinguishable latent states.

**Questions:**

- The authors mention in their introduction (end of second paragraph) that “[hybrid first-principles data-driven models] focus on state estimation using data-driven components to improve or augment existing dynamics but fail to learn global models and do not scale for large parameterized models. “. Can you provide some evidence or reference for these claims?

- The authors describe the process that governs the evolution of the parameters $\theta(t)$ and latent state $x$ as a Markov/diffusion process, but they nevertheless model it with an ODE. As I understand they do not refer to the Liouville formulation of the marginal density (probability flow ODE). I think something is amiss here. Can the authors explain or correct?

- The optimisation of model parameters and latent states for each time step, described in Eq. 12, is performed according to the Newton step outlined in Eqs. 14-16. How do you ensure convergence of these optimisation steps?

- Related to the previous question, isn't the overall method too time-consuming. As I understand, for each time step the method requires independent Newton updates until convergence. Can you provide some results discussing the computational complexity/compute requirements of the approach?

- In Section 4.2, please add a reference to the appendix where you detail how the estimation of the initial condition is performed (Appendix C).

- How does the proposed optimisation strategy relate to Expectation-Maximisation approaches for inference.

- Do you have any insight why the PR-SSM yields so unsatisfactory results?

- Do you have any insights on the learned vector fields? The presented experiments demonstrate that the proposed approach performs quite well for state estimation. Have you compared the learned vector fields to the ground truth ones?


- Minor: There are some typos in the main text and appendix (non capitalised letters, and missing articles).

---

> ### Author Response · Authors · 2023-11-22
> **Reply to the Weaknesses**
>
> Thank you very much for your deep understanding and valuable suggestions. We are pleased that you recognized the novelty and performance of the proposed approach.  First, we would like to highlight to the reviewer that, during testing, we did not perform parameter estimation, we learned the parameters, and the results displayed in the paper show the learned vector field with respect to the ground truth. The set of parameters found in the final training step $\hat{\theta}(t_N)$ are saved, then plugged into equation (4) of the original manuscript, which integrates the learned vector field and produces the plots corresponding to RNODE. We thought it was important to clarify this point because we felt it was not properly conveyed on our part and not properly understood by the reviewer.
>
> Bellow we reply to the Weaknesses point by point:
>
> - That is true, we thank you for pointing that out. We added the stochastic terms to our model description in sections 3.1 and 3.3 of the revised manuscript.  Added zero-mean noise, which makes our model equivalent to a stochastic differential equation in [4].
>
> - We thank the reviewer for bringing up this important point and we would like to take this opportunity to clarify our contribution further. Our framework learns dynamics governing unobserved states, and the only way these states can be identified is through their relationship with observed states.
> For this reason, unobserved states should be distinguishable. For example, let's assume we have a system of neural ODEs described as follows:
>
> $\dot{x}_1(t) =f_1(x_1(t),x_2(t),\theta_1)$
>
> $\dot{x}_2(t) =f_2(x_1(t),x_2(t),\theta_2)$
>
> $y(t)=h(x_1(t),x_2(t))$
>
> where our task is to learn $f_2(x_1(t),x_2(t),\theta_2)$ without measuring $x_2(t)$.
> Therefore, to identify $x_2(t)$, the system has to be state identifiable, that is $x_2(t)$ has to be recoverable from observations $y(t)$.
>
> A simple example of a undistinguishable scenario is $y(t) = h(x_1(t)=x_1^2(t)$. It is clear that in this scenario one cannot uniquely recover $x_1(t)$ from $y(t)$ since $x_1(t)$ is undistinguishable and has two possible values given one measurement $y(t)$.
>
> Another example of a distinguishable scenario is $\dot{x}_1(t)=x_2(t)$ and $y(t)=x(t)$. Assuming we observed $y(t)=x_1(t)$ at $t_0 $ and $t_1$, then $x_1(t_1)-x_1(t_0) =  (t_1-t_0)x_2(t_1)$,
> therefore $x_2(t_1)=\frac{x_1(t_1)-x_1(t_0)}{t_1-t_0}$ is the unique solution. It is clear that in this scenario one can uniquely recover $x_2$ from $y$  with knowledge of $t_1$ and $t_0$ since $x$ is distinguishable.
>
> We agree although, that distinguishability condition is sufficient to ensure identifiability but may not be necessary. In that sense, yes this could be quite limiting.
>
> To our knowledge, and as demonstrated by our experiments, there is no machine learning approach capable of learning ODEs governing unmeasured processes without requiring identifiability even if not stated.
>
> In control theory, necessary conditions for state identifiability are for classical systems is extensively covered under Observability of linear and nonlinear system.
> However, determining the necessary conditions for state identifiability in dynamical models with neural networks is still a very challenging open problem and is outside the scope of this paper. Therefore, we still do not have the necessary tools to identify under which scenario distinguishability condition could be eased.

---

> ### Author Response · Authors · 2023-11-22
> **Reply to the Questions**
>
> - These works focused on the filtering aspects where during training and testing parameters and states were continually updated. In RNODE, we aim at learning a global model. While we performed recursive joint estimation of states and parameters during the training phase, we only integrated the learned system during test. We also included a sentence in the third paragraph of Section~5 to highlight this point: "All results were obtained with learned mean vector field integrated over time."
>
> - That is true, we thank you for pointing that out. We corrected our model description sections 3.1 and 3.3 in the revised version where we refer to the model as a stochastic differential equation.
>
> - Since the optimization steps are performed according to Newton’s second-order method, they enjoy the same convergence properties as Newton’s method for every optimization step.
>
> - We agree that a complexity analysis should be presented, for this reason, we added a complexity section in the appendix. The complexity of the overall method at each epoch is $\mathcal{O}(N(2d_{\theta}^2d_x + 2d_{\theta}d_x^2))$, which is time-expensive if compared to gradient methods such as SGD and Adam. We would like to reiterate that computation speed was not the core focus of the work, instead, we targeted learning under partial observation. We would also like to point also that the method can be improved by Hessian approximations and other second-order optimization approximations used in [1], [2], and [3]. Nevertheless, that was not the focus of our work and we leave it for future work.
>
> - Thank you for pointing that out, we added the reference.
>
> - RNODE is a sequential joint estimation procedure for learning systems dynamics. Our understating is that the proposed approach is not directly connected to Expectation-Maximization (EM) in the sense that we are not maximizing a lower-bound of the marginal likelihood.
>
> - We emphasized in section 5 of the paper that PR-SSM does not account for the ODE structure and replaces all of the ODE dimensions with a neural network. Moreover modifying the PR-SSM method to account for ODE structure was out of the scope of the paper.
> We suspect that since all the dimensions are modeled as neural networks, the hidden state is not recoverable since the neural networks used are not state identifiable, which is a requirement to learn hidden state dynamics as we explained in the previous reply. Therefore, incorrect values of the hidden state are used in parameter inference.
>
>
>
> -  During testing, we did not perform parameter estimation. All results, in Section 5 and the appendix, were performed by simply integrating the \textit{learned} vector fields which were compared with the ground-truth. The set of parameters found in the final step, $\hat{\theta}(t_N)$, are saved and then plugged into equation (4) of the original manuscript, which integrates the \textit{learned} vector field and produces the plots and nRMSEs corresponding to RNODE. To clarify this in the manuscript we include the sentence "All results were obtained with learned mean vector field integrated over time." in the third paragraph of Section~5.
>
> - We revised the manuscript for typos.
>
> [1] Gupta, Vineet, Tomer Koren, and Yoram Singer. "Shampoo: Preconditioned stochastic tensor optimization." International Conference on Machine Learning. PMLR, 2018.
>
> [2] Anil, Rohan, et al. "Scalable second order optimization for deep learning." arXiv preprint arXiv:2002.09018 (2020).
>
> [3] Peirson, Abel, et al. "Fishy: Layerwise Fisher Approximation for Higher-order Neural Network Optimization." Has it Trained Yet? NeurIPS 2022 Workshop. 2022.
>
> [4] Øksendal, B. and Øksendal, B., 2003. Stochastic differential equations (pp. 65-84). Springer Berlin Heidelberg.

---

> > ### Comment · Reviewer_gkPz · 2023-11-22
> >
> > Thank you very much for the detailed response and the clarifications!
> >
> > >Thank you very much for your deep understanding and valuable suggestions. We are pleased that you recognized the novelty and performance of the proposed approach. First, we would like to highlight to the reviewer that, during testing, we did not perform parameter estimation, we learned the parameters, and the results displayed in the paper show the learned vector field with respect to the ground truth. The set of parameters found in the final training step
> > are saved, then plugged into equation (4) of the original manuscript, which integrates the learned vector field and produces the plots corresponding to RNODE.
> >
> > >During testing, we did not perform parameter estimation. All results, in Section 5 and the appendix, were performed by simply integrating the \textit{learned} vector fields which were compared with the ground-truth. The set of parameters found in the final step, are saved and then plugged into equation (4) of the original manuscript, which integrates the \textit{learned} vector field and produces the plots and nRMSEs corresponding to RNODE. To clarify this in the manuscript we include the sentence "All results were obtained with learned mean vector field integrated over time." in the third paragraph of Section~5.
> >
> > I think there might be a misunderstanding here. I understand that you didn't learn model parameters, since you extended the NODE formalism to learn the observed system. However, the figures in the manuscript show that your framework performs well in generating trajectories that resemble the ground truth ones (state estimation). However, you do not compare the actual underlying vector fields. Since your framework considers input driven dynamics (the data you consider for learning are pairs of inputs and observed states), it could be that the inferred vector fields by RNODE do not necessarily match the ground truth one, but the external inputs dominate the observed behavior, overshadowing the differences in internal dynamics.
> > This question was meant as a sanity check.
> >
> > Regarding the rest of the responses, thank you again very much for the clarifications.
> > I still cannot see the revised manuscript, so I will update my rating once this becomes accessible.

---

> ### Author Response · Authors · 2023-11-23
> **Vector field comparison**
>
> There was indeed a confusion. We tried to address this point in the last few hours by including plots of the true and estimated vector fields for the Harmonic Oscillator example in Appendix B4 (see Fig. 9).
> The revised manuscript has been submitted. We thank the reviewer one more time for the valuable comments.

---

### Official Review · Reviewer_W36z · 2023-11-06

**Soundness:** 3 good
**Presentation:** 4 excellent
**Contribution:** 3 good
**Rating:** 6
**Confidence:** 4

**Summary:**

In the present work the authors present a new extension to the Neural ODE approach centered around a recursive two-stage sequential optimization algorithm utilizing second-order information to be able to avoid vanishing gradients, but also avoids the pitfall of neural ODEs of optimizing all states at the same time by adopting the recursive, sequential optimization procedure.

The approach is validated across 5 examples, and compared to the other state estimation techniques of Buisson-Fenet et al. (RM), and Doerr et al. (PR-SSM).

**Strengths:**

The paper shines in its clarity, and the quality of its technical derivations, which are supplemented by a set of challenging experiments with the cart-pole, the harmonic oscillator, and the electro-mechanical positioning system. Especially the core component of the proposed algorithm, the sequential Newton procedure, is very well derived, and hence makes the paper's contributions very clear.

**Weaknesses:**

While strong on the technical side the paper at times lacks connection to the wider modern literature. This shows specifically in the section on 2nd-order optimizers, which lack acknowledgement of modern 2nd-order approaches such as Shampoo [1], Distributed Shampoo [2] and Fishy [3]. At the same time improved neural ODE algorithms like e.g. heavy-ball Neural ODEs [4] are not considered, and it would help improve the paper if the authors would set RNODE better in relation to existing literature in that regard.

In addition, there is some confusion with regards to the positioning of the algorithm present in the paper. Is it the goal to avoid the simultaneous estimation of all states by replacing it with the recursive approach, or avoid the vanishing gradients? This unclarity is present throughout the draft, and it would improve the paper greatly to clarify the focus of the algorithm, and the present weaknesses of existing approaches it addresses. Especially the claims that the RNODE approach avoids vanishing gradients would be helped by benchmarking RNODE against existing NODE-approaches, or for example the original version, on examples where classical NODEs suffer from vanishing gradients, and RNODE should then still be able to learn.

A further source of weakness of the paper is its experimental evaluation. While nominally having sufficient experimental evaluation with
- Neuron model
- Retinal circulation
- Cart-pole
- Harmonic oscillator
- EMPS
The first two benchmarks, i.e. the neuron model, and the retinal circulation are of limited information value as the results are all close together, and it would require error bars for e.g. 20 runs around the results to discern if one approach is actually better than the others in this instance. I believe there are a number of ways this weakness could be addressed:
a) Report error-bars on all experiments
b) Cut out the first two experiments, and replace them with a more difficult example such as the yeast glycosis of [5].
In addition there is no comparison to existing NODE approaches. Adding the vanilla NODE of Duvenaud et al. to the evaluation would help to add further context here.

[1] Gupta, Vineet, Tomer Koren, and Yoram Singer. "Shampoo: Preconditioned stochastic tensor optimization." International Conference on Machine Learning. PMLR, 2018.
[2] Anil, Rohan, et al. "Scalable second order optimization for deep learning." arXiv preprint arXiv:2002.09018 (2020).
[3] Peirson, Abel, et al. "Fishy: Layerwise Fisher Approximation for Higher-order Neural Network Optimization." Has it Trained Yet? NeurIPS 2022 Workshop. 2022.
[4] Xia, Hedi, et al. "Heavy ball neural ordinary differential equations." Advances in Neural Information Processing Systems 34 (2021): 18646-18659.
[5] Kaheman, Kadierdan, J. Nathan Kutz, and Steven L. Brunton. "SINDy-PI: a robust algorithm for parallel implicit sparse identification of nonlinear dynamics." Proceedings of the Royal Society A 476.2242 (2020): 20200279.

**Questions:**

* How do the authors see their approach scale to higher-dimensional dynamical systems?
* What would be a practical example of RNODE estimating the wrong state (page 4, end of 1st paragraph), and in turn finding the wrong model parameters? Is there a practical example, where such issue could occur?
* What are the costs of the _Sequential Newton_ optimization approach? To give a more complete picture here, the overall computational cost on a CPU or GPU, with the additional measurement of the # of function evaluations would help greatly to shed more light here.
* Have the authors validated their claim, that the gradient does not vanish with RNODEs?
* How do you see your approach scale to larger-scale data assimilation problems?

---

> ### Author Response · Authors · 2023-11-22
> **Reply to the Weaknesses**
>
> Thank you very much for your detailed review. We are very pleased that the reviewer find our contributions clear and our mathematical derivation of high quality. We thank you also for highlighting our paper strength and weakness which made us improve the paper’s quality significantly.
>
> Below is reply to the weaknesses point by point :
>
> - We thank the reviewer for pointing out these valuable references which we included in the revised manuscript in our related work section.  We would like to highlight that in our literature review we focused on articles that tackled the partially observed systems which poses additional challenges to the learning process, overlooking speeding up second order methods since that is not the main focus of our work. Nevertheless, we believe addressing this complexity issue is of very high importance to be addressed in future work. Moreover, NODE extensions such as Heavy ball NODE [4] deal with irregularly sampled time-series but assumes that the system is fully observed. For this reason we only included this reference in the Introduction.
>
> -  We agree with the reviewer that the positioning of the algorithm should be clarified.
> RNODE is a sequential (recursive) optimization approach that at each step solves an alternating optimization problem for learning system dynamics under partially observed states.
> The benefit of the sequential strategy is twofold:
> $(1)$ reduce the need for accurate initial conditions during training;
> $(2)$ avoids simultaneous estimation of all states, making second-order optimization methods feasible.
> Similarly, the alternating optimization approach improves the optimization of system parameters since it estimates unobserved hidden states and uses them in learning system parameters. In the case of RNODE, it also prevents vanishing gradients.
> To make these points clearer in the manuscript we modified the last paragraph of the introduction.
> Regarding the NODE, in the proposed method we do not use adjoints and having a dedicated section for this end made the positioning of the paper confusing. For this reason, we removed section 3.3 from the revised manuscript.
>
> - We agree with the reviewer that the neuron model and the retinal circulation results are close together. We tried to highlight that RNODE shows better performance in terms of RMSE (Table 1).
> We understand that adding confidence intervals would be beneficial, but we highlight that in our experiments and results, we are only integrating the mean learned dynamics. That is why we did not include uncertainty measures in the plots.
> Regarding the complexity of examples, although we believe the neuron model is fairly complex. We also agree that adding a more difficult yeast glycolysis example would strengthen the paper. We included this new experiment in the appendix of the revised manuscript. Moreover, the reviewer pointed out that there is no comparison to existing NODE approaches, and we added the vanilla ODE to our experimental benchmarks as mentioned in the previous reply.

---

> ### Author Response · Authors · 2023-11-22
> **Reply to the questions**
>
> Below is the reply to the questions point by point:
>
> - In the proposed approach, the network parameters are included as part of the state space. Therefore, since the size of the neural network is already way larger than the system dimension, the proposed approach would not have a problem with moderate dimensional systems since it would not add many variables to the state space compared to the number of neural network parameters. However, our  approach may scale badly to very higher-dimensional neural network architectures since it’s a second-order Newton method at its core.
> To calculate the complexity of our algorithm,
> first, we need to calculate the complexity of each Jacobian. Since we are using automatic differentiation to calculate them, the complexity of getting $F_{x_{i-1}}, G_{\theta_{i-1}}$, and $F_{\theta_{i-1}}$ is $\mathcal{O}(d_x^2),  \mathcal{O}(d_{\theta}^2)$ and $\mathcal{O}(d_{\theta}d_x)$, respectively.
> However, the complexity of $G_{\theta_{i-1}}$ could be removed if $g(\theta(t_i))=0$ since $G_{\theta_{i-1}}=I$ in that case. Note that this is the case in our experiments.
> Assuming $n_x \approx n_y$, the complexity of calculating  $P_{\theta_i}^{-}$,  $P_{x_i}^{-}$, $P_{x_i}$  and $P_{\theta_i}$ are $\mathcal{O}(2d_{\theta}^2d_x + d_x+ (2d_x^2 + 1)d_{\theta})$, $\mathcal{O}(2d_x^3 + d_x)$, $\mathcal{O}(4d_x^3 + 2d_x) $ and $ \mathcal{O}(d_\theta)$, respectively.
> Moreover, the complexity of computing $\hat{x}(t_i)$ is $\mathcal{O}(2d_x^3+d_x^2)$, and the complexity of $\hat{\theta}(t_i)$ is $\mathcal{O}(d_{\theta}(d_x+1))$.
> Since the Jacobians are differentiated once, then evaluated at each time step, the complexity of the whole algorithm with a dataset of size $N$ for each epoch becomes:
> \begin{equation}
>     \mathcal{O}(N(d_x^3 + d_\theta^2 d_x + d_x^2 d_\theta)).
> \end{equation}
> Finally, assuming $d_{\theta} \gg d_x$, the total cost of each training epoch can be simplified as:
> \begin{equation}
>     \mathcal{O}(N(2d_{\theta}^2d_x + 2d_{\theta}d_x^2))
> \end{equation}
> It is clear that the main source of complexity would be $d_{\theta}$.
> Thus, RNODE may scale badly to very higher-dimensional neural network architectures.
> However, fast approximations of Newton's method exist, as pointed out by the reviewer, such as Shampoo. Although merging Shampoo with our proposed approach could reduce the computational burden, we will analyze this hypothesis in future works.
>
> - A practical example of RNODE estimating the wrong state and turn in finding the wrong parameters would be the following example. let's assume we have a system of neural ODEs described as follows:
> $$ \dot{\theta}(t)=g(\theta(t))$$
> $$\dot{x}_1(t)=x_2^2(t)$$
> $$\dot{x}_2(t)=f_2(x_1(t),x_2(t),\theta(t))$$
> $$y(t)=x_1(t)$$
> where our goal is to learn $f_2(x_1(t),x_2(t),\theta(t))$ with $y=x_1(t)$. Assuming we observed $y=x_1(t)$ at $t_1 $ and $t_2$, then $x_1(t_1)=\frac{x_1(t_1)-x_1(t_0)}{t_1-t_0}=x_2^2(t_1)$
> therefore $x_2(t_1)=\pm\sqrt{\frac{x_1(t_1)-x_1(t_0)}{t_1-t_0}}$. It is clear that in this scenario one cannot uniquely recover $x_2(t)$ from $y=x_1(t)$  since $x_2(t)$ is not distinguishable and has two possible solutions.\\
> Therefore, $\theta(t)$ would be learned according to equation (18) in the paper using a wrong value of $x_2$ and would result in learning the wrong vector field $f_2(x_1(t),x_2(t),\theta(t))$.
>
> - The computation complexity of sequential Newton optimization is $\mathcal{O}(N(2d_{\theta}^2d_x + 2d_{\theta}d_x^2))$ per each epoch as calculated above.
>
> - Yes, we have validated the claim that gradients do not vanish with the proposed method RNODE since we were able to learn the hidden dynamics.
>
> - In large-scale data assimilation problems $d_x$ is often very-large. In this context, the proposed approach will suffer during training since large neural networks and number of states are present.
> In our experiments, we only integrated the learned mean vector field during testing. In this scenario, the complexity at each time step is fairly small (just the application of the dynamical model).
> If data assimilation is performed with respect to the states but keeping the learned parameters $\theta$ constant, the cost will be as follows assuming $d_x \approx d_y$:
>       $$  \mathcal{O}(d_{x}^2)  + N[\mathcal{O}(2d_x^3 + d_x) + \mathcal{O}(4d_x^3 + 2d_x) + \mathcal{O}(2d_x^3+d_x^2) ]  $$ which is equivalent to $\mathcal{O}(Nd_x^3)$ for the whole dataset of size N. However, if the learning is also performed during assimilation periods, which can be beneficial  in non-stationary scenarios, then the cost will be similar to the training, in which case the method will suffer due to large dimensional states and parameters.

---

> > ### Author Response · Authors · 2023-11-22
> > **References**
> >
> > References:
> >
> > [1] Gupta, Vineet, Tomer Koren, and Yoram Singer. "Shampoo: Preconditioned stochastic tensor optimization." International Conference on Machine Learning. PMLR, 2018.
> >
> > [2] Anil, Rohan, et al. "Scalable second order optimization for deep learning." arXiv preprint arXiv:2002.09018 (2020).
> >
> > [3] Peirson, Abel, et al. "Fishy: Layerwise Fisher Approximation for Higher-order Neural Network Optimization." Has it Trained Yet? NeurIPS 2022 Workshop. 2022.

---

> ### Author Response · Authors · 2023-11-23
> **Revised manuscript submitted**
>
> Dear reviewer,
>
> we submitted the revised manuscript. We believe to have addressed most of the reviewer's points and hope the revised manuscript matches your expectations. Thank you for your valuable input.

---

> > ### Comment · Reviewer_W36z · 2023-11-23
> > **Thank you for addressing the concerns**
> >
> > I sincerely thank the authors for addressing my, and the other reviewers' concerns. I believe the improvements have made the paper significantly more clear, and the additional experiments have also made the approach better comparable to state-of-the-art approaches.